# Tunable chiral and nematic states in the triple-Q antiferromagnet $Co_{1/3}TaS_2$

Erik Kirstein [1,7], Pyeongjae Park [2,3,7], Woonghee Cho[3], Cristian D. Batista [4,5] ✉, Je-Geun Park [3,6] ✉ & Scott A. Crooker [1] ✉

Complex spin configurations in magnetic materials, ranging from collinear single-$Q$ to non-coplanar multi-$Q$ states, exhibit rich symmetry and chiral properties. However, their detailed characterization is often hindered by the limited spatial resolution of neutron diffraction techniques. Here we employ magnetic circular dichroism and magnetic linear dichroism to investigate the triangular lattice antiferromagnet $Co_{1/3}TaS_2$, revealing three-state ($Z_3$) nematicity and also spin chirality across its multi-$Q$ magnetic phases. At intermediate temperatures, the presence of linear dichroism identifies nematicity arising from a single-$Q$ stripe phase, while at high magnetic fields and low temperatures, a phase characterized solely by circular dichroism emerges, signifying a purely chiral non-coplanar triple-$Q$ state. Notably, at low temperatures and small fields, we discover a unique phase where both chirality *and* nematicity coexist. A theoretical analysis based on a continuous multi-$Q$ manifold captures the emergence of these distinct magnetic phases, as a result of the interplay between four-spin interactions and weak magnetic anisotropy. Additionally, both circular and linear dichroism microscopy spatially resolves the chiral and nematic domains. Our findings establish $Co_{1/3}TaS_2$ as a rare platform hosting diverse multi-$Q$ states with distinct combinations of spin chirality and nematicity while demonstrating the effectiveness of polarized optical techniques in characterizing complex magnetic textures.

Antiferromagnets, long overshadowed by their ferromagnetic counterparts, are rapidly emerging as a new frontier in condensed matter physics[1–4]. Their complex spin orders offer deep insights into magnetic symmetries, chiralities, and associated emergent phenomena[5–8]. Antiferromagnets (AFMs) can exhibit diverse spin configurations, ranging from collinear structures described by a single ordering wave vector $Q$, to complex non-collinear and non-coplanar multi-$Q$ spin configurations arising from the superposition of multiple Fourier components with distinct wave vectors $Q_\nu$ ($\nu = 1, 2, 3..,$). Notably, multi-$Q$ states can give rise to topologically non-trivial spin textures[9–12] that manifest

unique phenomena[13–17]. Understanding these AFM orders requires precise identification of their magnetic symmetries and potential topological properties, both of which are central themes in modern magnetism research.

The triangular lattice antiferromagnetism in $Co_{1/3}TaS_2$, an intercalated metallic van der Waals system with $Co^{2+}$ spins on 2D triangular lattices (depicted in Fig. 1a), provides an excellent platform for exploring the rich physics associated with competing AFM orders. In this system, conduction-electron-mediated two-spin and four-spin interactions stabilize a non-coplanar triple-$Q$ ground state below

[1]National High Magnetic Field Laboratory, Los Alamos National Lab, Los Alamos, NM, USA. [2]Materials Science and Technology Division, Oak Ridge National Laboratory, Oak Ridge, TN, USA. [3]Department of Physics and Astronomy, Seoul National University, Seoul, Korea. [4]Department of Physics and Astronomy, University of Tennessee, Knoxville, TN, USA. [5]Shull Wollan Center—A Joint Institute for Neutron Sciences, Oak Ridge National Laboratory, Oak Ridge, TN, USA. [6]Institute of Applied Physics, Seoul National University, Seoul, Korea. [7]These authors contributed equally: Erik Kirstein, Pyeongjae Park. ✉e-mail: cbatist2@utk.edu; jgpark10@snu.ac.kr; crooker@lanl.gov

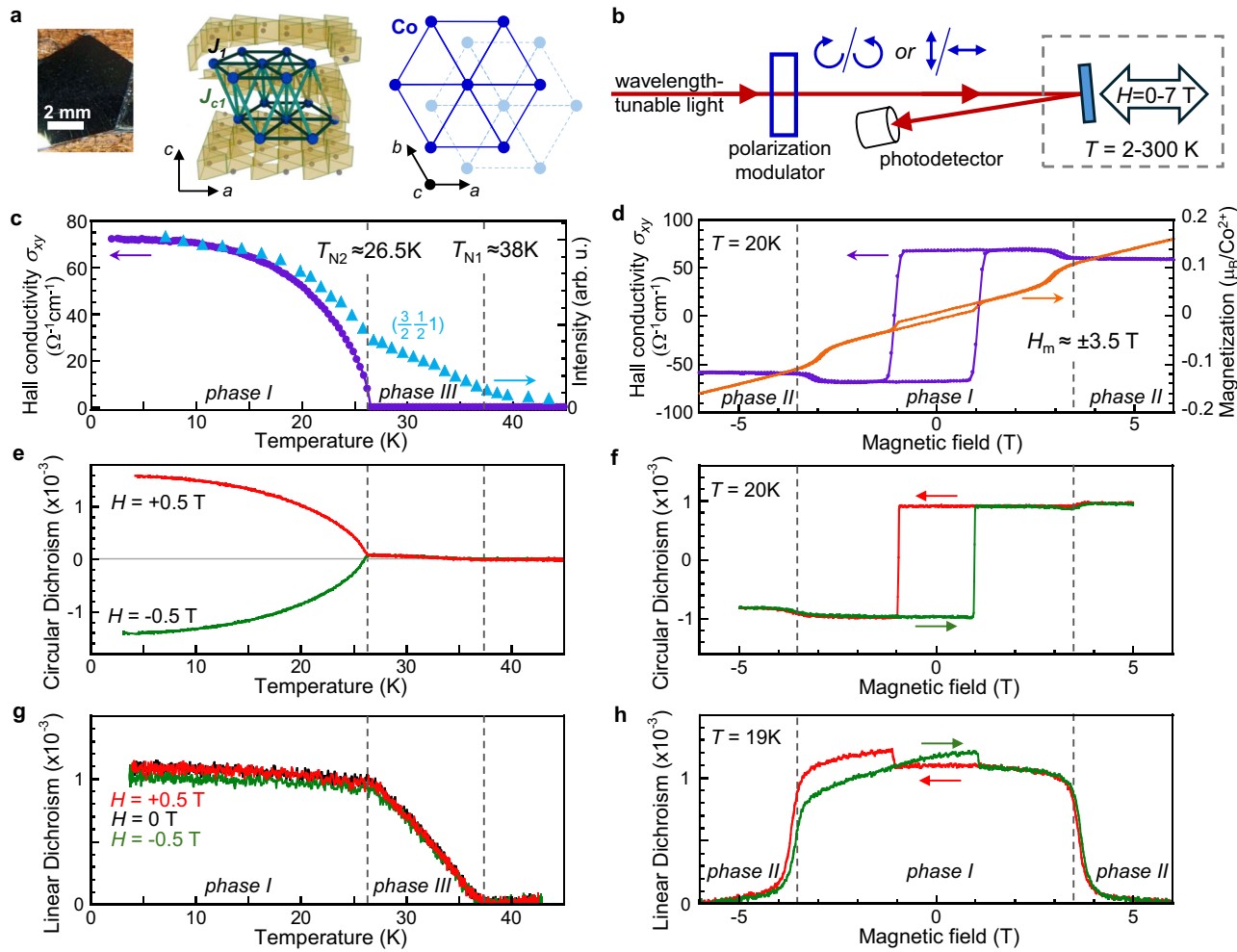

**Fig. 1 | Optical detection of chiral and nematic antiferromagnetic order in Co$_{1/3}$TaS$_2$. a** Illustration of the Co$_{1/3}$TaS$_2$ crystal structure. Co ions (blue) intercalate between TaS$_2$ monolayers, forming layers of spins on 2D triangular lattices with ABAB stacking, resulting in a tetrahedral network of Co spins with intra- and inter-layer exchange couplings $J_1$ and $J_{c1}$. **b** Schematic of the magnetic circular dichroism (MCD) and magnetic linear dichroism (MLD) experiment. Wavelength-tunable light is modulated between right- and left-circular polarization (for MCD) or between linear and cross-linear polarization (for MLD) by a photoelastic modulator, and then reflected from the sample at near-normal incidence and detected by a pho-todiode. Out-of-plane magnetic fields $H$ to ±7 T can be applied. MCD measures the normalized reflected intensity difference between right- and left-polarized light, $(I_R - I_L)/(I_R + I_L)$, and is sensitive to, e.g., chiral (triple-**Q**) AFM order. MLD measures the normalized reflected intensity difference between linear- and cross-linear light, $(I_\phi - I_{\phi+90°})/(I_\phi + I_{\phi+90°})$, and is sensitive to, e.g., nematic (stripe-like, single-**Q**) AFM order with broken in-plane ($C_{2z}$) symmetry. **c** The recently-reported Hall

conductivity $\sigma_{xy}$ and neutron diffraction intensity from Co$_{1/3}$TaS$_2$ vs. $T$ (from[19]), showing the onset of single-**Q** order below $T_{N1} = 38$ K and triple-**Q** (chiral) order below $T_{N2} = 26.5$ K. **d** The recently-reported $\sigma_{xy}$ and magnetization $M$ vs. $H$, showing large hysteretic $\sigma_{xy}$ despite small $M$, and additional metamagnetic transitions at $H_m \approx \pm 3.5$ T whose nature is not known to date. **e, f** MCD studies vs. $T$ and $H$ confirm the emergence of chiral AFM order below $T_{N2}$, and a large $H$-dependent hysteresis, closely following the electrical transport measurements of $\sigma_{xy}$ shown in the panels above. **g, h** MLD studies vs. $T$ and $H$ reveal the emergence of nematic order below $T_{N1}$ and its saturation below $T_{N2}$, and reveal that this nematicity exists only at low fields $|H| < H_m$. (Small linear backgrounds have been removed from $H$-dependent data of $\sigma_{xy}$, MCD, and MLD). These data point to nematic (single-**Q**) AFM order at high $T$, purely chiral ("three-fold symmetric triple-**Q**") AFM order at low $T$ and high $H$, and to the coexistence of both chiral *and* nematic ("distorted triple-**Q**") AFM order at low $T$ and low $H$. All optical data acquired using 650 nm light.

26.5 K[18,19]. This state, characterized by ordering wave vectors at the three $M$ points (edges) of the hexagonal Brillouin zone, represents the shortest-wavelength limit of a magnetic skyrmion crystal[20]. The resulting chiral spin texture can generate a pronounced topological Hall effect (without requiring relativistic spin-orbit coupling), leading to a large Hall conductivity $\sigma_{xy}$ despite vanishing magnetization, and even in the absence of applied magnetic field[18,19]. At intermediate temperatures between 26.5 K and 38 K, Co$_{1/3}$TaS$_2$ transitions to a stripe-like single-**Q** antiferromagnetic phase[19,21], which has been pro-posed as a promising candidate for realizing discrete three-state ($Z_3$) electronic nematicity[22–24], arising from rotational symmetry breaking in a hexagonal lattice. Beyond these zero-field chiral and nematic phases, out-of-plane magnetic fields $H \gtrsim 3.5$ T induce two additional

AFM phases, whose spin configurations and ordering mechanisms remain unknown[18,19,25].

These recent discoveries raise fundamental questions about how magnetic chirality and nematicity evolve across these temperature- and field-induced phases. Prior studies of similar phenomena are scarce[26,27], and neutron diffraction methods applied to date struggle to fully distinguish and characterize magnetic chirality and nematicity, due to a lack of spatial resolution and inability to resolve magnetic domains[18,19]. This challenge highlights the need for complementary approaches to investigate the rich ($H$, $T$) phase diagram of Co$_{1/3}$TaS$_2$, ideally with experimental probes that can illuminate the interplay and potential coexistence of single-**Q** and triple-**Q** orderings within the same system.

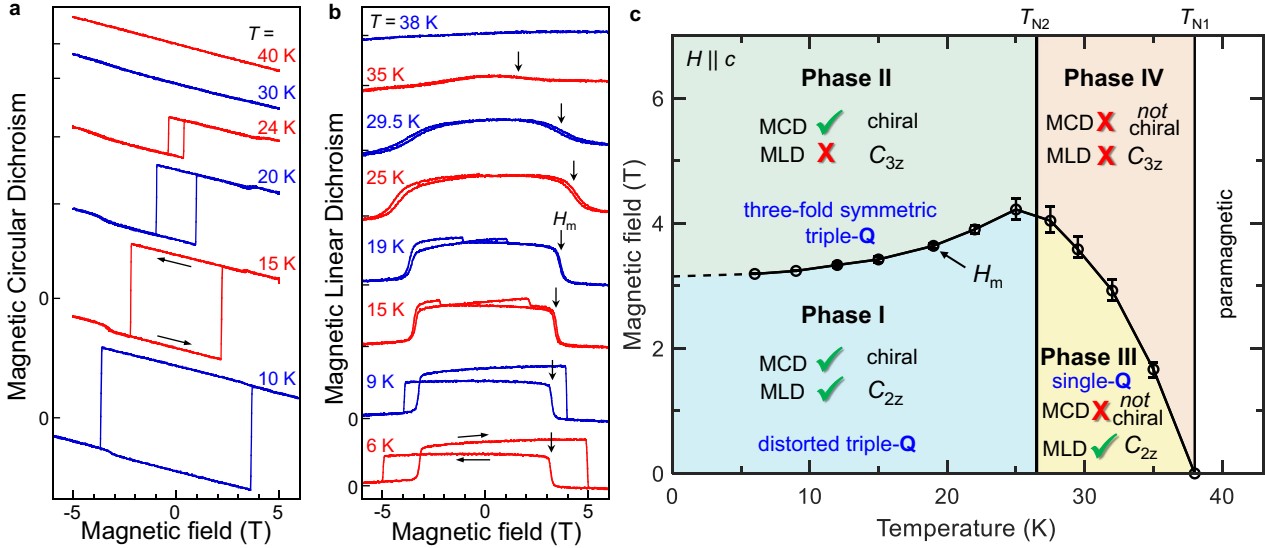

**Fig. 2 | Mapping out the antiferromagnetic phase diagram of $Co_{1/3}TaS_2$.**
**a, b** MCD and MLD versus $H$, at different temperatures, showing how signatures of chiral and nematic AFM order, revealed by MCD and MLD respectively, emerge and/or vanish with changing $T$ and $H$ (curves offset for clarity). All data acquired using $\lambda = 650$ nm. **c** Magnetic phase diagram of $Co_{1/3}TaS_2$ based on these optical data, where the four different Phases I-IV are defined by the presence or absence of spontaneous topological MCD (chiral AFM order), and MLD (nematic, stripe-like AFM order).

Here we demonstrate that optical methods for magnetic circular dichroism (MCD) and magnetic linear dichroism (MLD) directly provide spatially-resolved measurements of chirality and nematic symmetries, respectively, of the different antiferromagnetic orders in $Co_{1/3}TaS_2$. These techniques reveal stripe-like ($Z_3$ nematic) antiferromagnetism arising from single-**Q** order at intermediate $T$, purely chiral order ("three-fold symmetric triple-**Q**") at low $T$ and large $H$, and – crucially – the coexistence of *both* chiral and nematic ("distorted triple-**Q**") order at low $T$ and $H$. Our findings establish $Co_{1/3}TaS_2$ as a rare material hosting diverse and tunable multi-**Q** states, and highlight the power of magneto-optical techniques for characterizing complex magnetic textures.

## Results

### Antiferromagnetic phases in $Co_{1/3}TaS_2$

$Co_{1/3}TaS_2$ consists of 2D triangular lattices of Co spins, intercalated within the van der Waals gaps of $2H$-$TaS_2$ (see Fig. 1a). Nearest-neighbor intra- and inter-layer couplings and AB stacking create an effective tetrahedral network of Co, which orders antiferromagnetically. Figure 1c, d briefly review its diverse AFM phases, revealed by recent transport, neutron scattering, and magnetization studies[18,19,25]. When cooled in $H = 0$, AFM order first appears below $T_{N1} = 38$ K, where neutron studies identify collinear single-**Q** (stripe) order[21] with Co spins aligned out-of-plane[18,19]. Upon further cooling, a new AFM ground state emerges below $T_{N2} = 26.5$ K that is characterized by a large spontaneous Hall conductivity $\sigma_{xy}$, despite vanishingly small zero-field magnetization ($M_z = 0.01\ \mu_B/Co^{2+}$). This low-temperature phase is identified as a chiral non-coplanar "tetrahedral" triple-**Q** order. Electrons moving within this chiral spin texture accumulate a geometric (Berry) phase that generates an emergent magnetic field, resulting in substantial $\sigma_{xy}$ despite tiny $M_z$ – generally referred to as a topological Hall effect[13,28–30]. These triple-**Q** and single-**Q** phases are designated Phases I and III, respectively[18].

Crucially, applying out-of-plane $H$ larger than $H_m \approx 3.5$ T) induces additional metamagnetic transitions, observed as small jumps in $\sigma_{xy}$ and $M_z$ (see Fig. 1d)[18,25]. These transitions lead to two additional phases —Phase II and Phase IV—presumably with distinct spin configurations and ordering mechanisms that remain unresolved (note that the data shown in Fig. 1c, d do not explicitly reveal the transition to Phase IV; however, this transition is shown in refs. 18,19,25, and is also evident in Fig. 2 below).

To directly probe the magnetic order of these phases, we employ MCD and MLD – optical techniques that can selectively measure chiral and nematic properties, respectively. The experiment is depicted in Fig. 1b (see caption for details, and Methods). Traditionally, MCD has been used to detect ferromagnetic order, as it probes the nonzero off-diagonal optical conductivity $\sigma_{xy}(\omega)$, which also underpins the dc Hall effect at $\omega \approx 0$. However, recent discoveries of antiferromagnets with non-trivial spin orders whose (reduced) symmetry allows for $\sigma_{xy}(\omega)$ – such as non-collinear $Mn_3Sn$[31–34] or here for non-coplanar $Co_{1/3}TaS_2$ – suggests that MCD can directly probe such AFM order[35–39].

Complementing MCD, MLD has recently been demonstrated as an effective probe of stripe-like single-**Q** AFM order in 2D hexagonal magnets[22–24]. Such order breaks the rotational symmetry of the underlying crystal, inducing in-plane ($C_{2z}$) anisotropy of the optical conductivity and giving rise to three-state $Z_3$ nematic order (note: by $C_{2z}$ we mean the subgroup consisting of the identity and the transformation $(x, y, z) \rightarrow (-x, -y, z + \frac{1}{2})$, which is the product of a $\pi$ rotation about the $c$-axis and translation along the same axis). Together, MLD and MCD provide a powerful framework for characterizing multi-**Q** magnetic states in $Co_{1/3}TaS_2$.

### Detecting chiral and nematic AFM order with MCD and MLD

To establish MCD as a probe of the chiral triple-**Q** order, we measure its temperature and field dependence. Figure 1e shows that strong MCD signals emerge below $T_{N2}$, closely following the behavior of $\sigma_{xy}(T)$ from transport measurements (Fig. 1c). Furthermore, Fig. 1f shows that MCD exhibits a marked hysteresis with $H$, in agreement with prior transport studies shown in Fig. 1d[18,19,25]. These findings confirm that MCD is sensitive to chiral AFM order in $Co_{1/3}TaS_2$. The scalar spin chirality of this triple-**Q** state, defined as $\chi_{ijk} = S_i \cdot (S_j \times S_k)$ where $i, j, k$ label spins around any triangular plaquette in the lattice, can be switched between time-reversed configurations by $H$ ($\approx \pm 1$ T at 20 K), giving opposite $\sigma_{xy}(\omega)$ and MCD. The sharp switching behavior indicates high sample quality and uniform reversal of the chiral

order. Additionally, MCD signals reveal the meta-magnetic transition at $H_m \approx \pm 3.5$ T, in close correspondence with transport and magnetization measurements (Fig. 1d; see also Supplementary Figs. S1 and S2). The correspondence of MCD and Hall conductivity suggests their common origin: the real-space Berry curvature generated by chiral triple-**Q** magnetism[18,19].

In marked contrast, Fig. 1g shows that MLD emerges below $T_{N1} = 38$ K, coinciding with the onset of stripe-like single-**Q** (nematic) AFM order. As shown in recent studies of 2D hexagonal antiferromagnets FePS$_3$ and Fe$_{1/3}$NbS$_2$[22-24], collinear single-**Q** stripe order can generate substantial MLD due to rotational symmetry breaking and consequent asymmetry of the in-plane optical conductivity. The MLD in Co$_{1/3}$TaS$_2$ saturates at a large value below $T_{N2}$, indicating the persistence of nematicity in the low-temperature phase. The MLD does not vary significantly when cooled in small $\pm H$, or in $H = 0$. Importantly, the simultaneous presence of both MLD and MCD at $T < T_{N2}$ (Fig. 1e, g) provides direct evidence that Phase I exhibits both chiral *and* nematic order.

Field-dependent MLD further clarifies the nature of the puzzling metamagnetic phase transition at $H_m \approx \pm 3.5$ T. As shown in Fig. 1h, the MLD is large at low $H$ but vanishes when $|H| > H_m$, indicating the *disappearance* of nematicity. Meanwhile, the MCD remains large (see Fig. 1f), demonstrating that the $H_m$ separates a nematic-chiral phase (Phase I) from a purely chiral phase (Phase II). Notably, small jumps in the MLD signal coincide with chirality reversals, likely arising in part from cross-talk between linear and circular dichroism signals. This cross-talk also generates the very small MCD signal between $T_{N1}$ and $T_{N2}$ in Fig. 1e (see Supplementary Fig. S2).

## Mapping the magnetic phase diagram

Figure 2a, b show field-dependent MCD and MLD at different temperatures. As previously observed in $\sigma_{xy}$[18,25], both the chiral switching field $H_c$ and the amplitude of the hysteresis loops increase rapidly below $T_{N2}$. Importantly, the disappearance of MLD at large $H$ persists for all $T < T_{N1}$, reinforcing the phase distinction [see also MLD($T$) scans at fixed $H$ in Supplementary Fig. S3]. Using these magneto-optical measurements, we construct the ($H$, $T$) phase diagram in Fig. 2c, defining Phases I-IV based on the presence or absence of MCD (chirality) and MLD (nematicity).

The MCD results confirm that Phases I and II exhibit chiral triple-**Q** order, while the high temperature Phases III and IV do not. More importantly, MLD offers new insights into the symmetries of Phases I, II, and IV. The absence of rotational $C_{3z}$ symmetry in Phase I, revealed by MLD, contradicts the recently-proposed threefold-symmetric triple-**Q** ground state[18]. Instead, the presence of both MCD and MLD in Phase I suggests a triple-**Q** state with *broken $C_{3z}$* rotational symmetry – that is, with both chiral *and* nematic AFM order. As described in the next section, such novel states can be realized through intermediate spin configurations involving coexisting single-**Q** and triple-**Q** components (related AFM phases were studied theoretically for Mn monolayers[26]). Finally, we emphasize that Phase II, which exhibits only MCD, appears to realize the $C_{3z}$-symmetric triple-**Q** state that was originally proposed for Phase I.

## Continuous multi-Q manifold model

These observations motivate a continuous multi-**Q** manifold model that smoothly interpolates between single-**Q** and three-fold symmetric triple-**Q** states, and which naturally captures the evolution of spin chirality and nematicity in Co$_{1/3}$TaS$_2$. For $M$-ordering wave vectors, this continuous interpolation can be modeled by the following manifold (see Fig. 3a):

$$\mathbf{S}(\mathbf{r}) = \sum_{\nu=1}^{3} \widetilde{\mathbf{S}}_{\mathbf{Q}_\nu} \cos(\mathbf{Q}_\nu \cdot \mathbf{r}), \qquad (1)$$

where $\mathbf{r}$ is a lattice vector on the 2D triangular lattice, and $\mathbf{S}(\mathbf{r})$ denotes a classical spin vector. The derivation of Eq. (1) and its validity for Co$_{1/3}$TaS$_2$ is described in Supplementary Section 2A. By adjusting the relative magnitudes of the three Fourier components ($|\widetilde{\mathbf{S}}_{\mathbf{Q}_\nu}| \equiv \Delta_\nu$) while maintaining their orthogonality and a constant total magnitude $\Delta_1^2 + \Delta_2^2 + \Delta_3^2$, this ansatz spans a spherical surface in ($\Delta_1, \Delta_2, \Delta_3$) phase space (see Fig. 3a), which interpolates between all possible single-, double-, and triple-**Q** spin configurations with fixed $|\mathbf{S}(\mathbf{r})|$. We also note that ref. 26 considered a 1D sub-manifold of the 2D manifold of degenerate ground states. However, Phase IV is not captured by Eq. (1) and will be discussed separately.

The case $\Delta_1 = \Delta_2 = \Delta_3$ (red circle in Fig. 3a) yields a chiral and three-fold symmetric ground state, where the four spin sublattices align along the principal axes of a regular tetrahedron with 109.5° mutual angles. Phase II corresponds to this case, but with the addition of a small net out-of-plane magnetization arising from spin canting induced by the applied magnetic field. This canted configuration is naturally described by including a constant (ferromagnetic) Fourier component $\widetilde{\mathbf{S}}_0$ to the three-dimensional manifold in Eq. (1); for additional details see Supplementary Section 2A and Eq. (S7). Importantly, even with canting, the three-fold symmetry remains intact (see Supplementary Fig. S6d) and therefore we refer to Phase II as a "three-fold symmetric triple-**Q**" state throughout this work, to emphasize its rotational symmetry (though not implying a perfect 109.5° tetrahedral configuration.). In contrast, a single non-zero $\Delta_\nu$ (blue circles in Fig. 3a) corresponds to the single-**Q** stripe order of Phase III (Fig. 3e). Intermediate states arise when $0 < \Delta_i = \Delta_j < \Delta_k$ for indices $i$, $j$, $k \in 1$, 2, 3 (green lines throughout Fig. 3). The broken $C_{3z}$ symmetry is evident from the unequal $\Delta_\nu$ magnitudes (Fig. 3c), resulting in the four spin sublattices spanning a *distorted* tetrahedron, which we term "distorted triple-**Q**". From this perspective, thermal fluctuations and $H$ cause the magnetic ground state to evolve smoothly within this multi-**Q** manifold, effectively controlling both the spin chirality and $Z_3$ nematicity in Co$_{1/3}$TaS$_2$.

While the emergence of a continuous multi-**Q** manifold is rarely observed in real systems, our theoretical analysis adds strong evidence for its feasibility based on a realistic spin Hamiltonian for Co$_{1/3}$TaS$_2$. The isotropic low-energy Hamiltonian introduced in previous studies[19,21], which incorporates bilinear Heisenberg and four-spin interactions, fails to fully capture the field- and temperature-driven multi-**Q** manifold. While a more generalized model could include additional four-spin terms with varying forms[40], we find that a simple real-space biquadratic (i.e., four-spin) term, $\widehat{\mathcal{H}}_{bq}$, captures the most general classical ground state of the multi-**Q** $M$-ordering and its long-wavelength fluctuations:

$$\widehat{\mathcal{H}}_{bq} = K \sum_{\mathbf{r}, \boldsymbol{\delta}_1} \left( \widehat{\mathbf{S}}_{\mathbf{r}} \cdot \widehat{\mathbf{S}}_{\mathbf{r}+\boldsymbol{\delta}_1} \right)^2, \qquad (2)$$

where $\boldsymbol{\delta}_1$ is the vector connecting nearest-neighbor sites (derived in Supplementary Sections 2B and C). Notably, $\widehat{\mathcal{H}}_{bq}$ with $K > 0$ successfully captures the single-**Q** to triple-**Q** transition at $T_{N2}$[19,21]. However, it incorrectly predicts a $C_{3z}$-symmetric triple-**Q** ground state at $T = H = 0$ (Phase I), and fails to explain any field-induced metamagnetic transition.

Magnetic anisotropy in Co$_{1/3}$TaS$_2$, though smaller than $K$ in magnitude, is suggested by the finite magnon energy gap in Phase I and by the out-of-plane spin configuration in Phase III[19,21] (see Supplementary Section 2D). To account for this, we include single-ion easy-axis anisotropy ($\widehat{\mathcal{H}}_{SI}$) and bond-dependent exchange anisotropy ($\widehat{\mathcal{H}}_{\pm\pm}$) in the

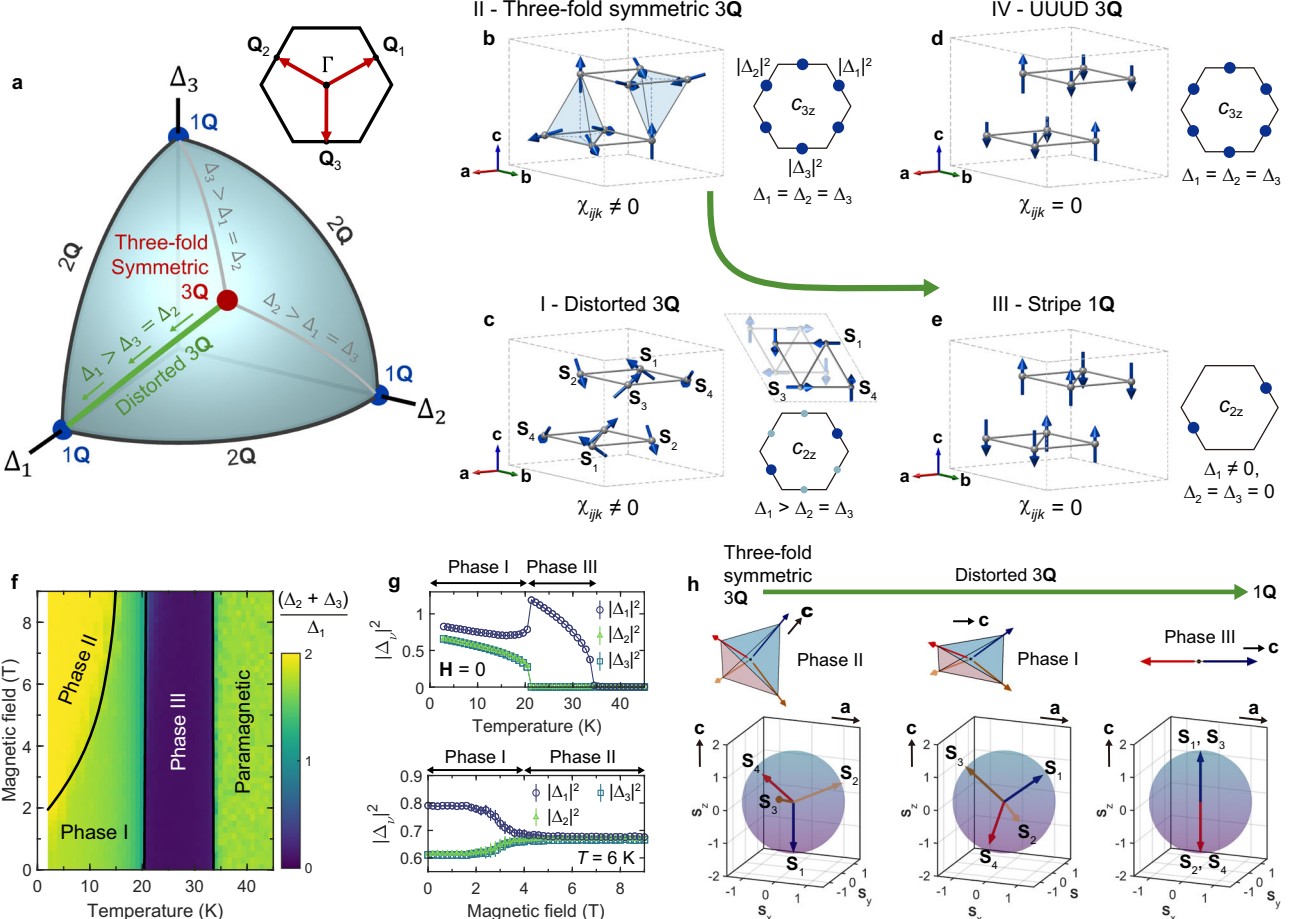

**Fig. 3 | Theoretical model of the $H$, $T$ phase diagram based on continuous multi-Q manifolds. a** Continuous manifold of multi-Q $M$-orderings, depicted as a variational space on a spherical shell. **b**–**e** Magnetic structures consistent with the symmetry and topological properties suggested by the combination of MCD and MLD measurements. The size and color of the blue dots on the right side of each panel represent the magnitude of the Fourier components ($|\Delta_\nu|^2$). **f**, **g** Theoretical temperature-field ($H \parallel c$) phase diagram derived from classical Monte-Carlo simulations of the realistic anisotropic spin model for $Co_{1/3}TaS_2$ (Eqs. (2) and (3)). The color code in **f** indicates $r_\Delta \equiv \frac{\Delta_2 + \Delta_3}{\Delta_1}$, where $\Delta_3 \leq \Delta_2 \leq \Delta_1$. Spin configurations of the Phases I–III are depicted in **c**, **b**, and **e**, respectively. **h** Evolution of the four spin-sublattice configurations throughout Phases II, I, and III, transitioning from a three-fold symmetric triple-Q (3Q) state to a stripe-like single-Q (1Q) state via a distorted tetrahedral alignment.

isotropic model:

$$\widehat{\mathcal{H}}_{SI} = A \sum_{\mathbf{r}} \left( \widetilde{S}_{\mathbf{r}}^z \right)^2,$$

$$\widehat{\mathcal{H}}_{\pm\pm} = \sum_{\langle i,j \rangle_1} 2J_{\pm\pm} \left[ \left( S_i^x S_j^x - S_i^y S_j^y \right) \cos\phi_\alpha \right. \tag{3}$$
$$\left. - \left( S_i^x S_j^y + S_i^y S_j^x \right) \sin\phi_\alpha \right],$$

where $\langle i, j \rangle_1$ runs over nearest-neighbor bonds, $x \parallel a$-axis, and $\phi_\alpha \in \{0, 2\pi/3, 4\pi/3\}$ is the angle between a bond vector $(i \to j)$ and the $a$-axis. The resultant phase diagram at $T = H = 0$ spanned by $J_{\pm\pm}$ and $A$ is shown in Supplementary Figs. S7b and S8, where we effectively visualize it through a quantity $r_\Delta \equiv \frac{\Delta_2 + \Delta_3}{\Delta_1}$, where $\Delta_3 \leq \Delta_2 \leq \Delta_1$. Introducing a non-zero $A$ immediately changes the ground state from a three-fold symmetric triple-Q ($r_\Delta = 2$, Fig. 3b) to a distorted triple-Q ($1 < r_\Delta < 2$, Fig. 3c), consistent with the observed MLD in Phase I. Thus, an intermediate triple-Q ground state on the continuous manifold is a natural explanation for Phase I under the presence of magnetic anisotropy.

Further exploration of temperature- and field-dependent ground states was performed using classical Monte Carlo simulations (see Methods, and Supplementary Section 2D). The results in Fig. 3f–h, presented as $\Delta_\nu$ and $r_\Delta$, reveal: (i) a distorted triple-Q ground state ($1 < r_\Delta < 2$) in Phase I, (ii) a field-induced three-fold symmetric triple-Q state ($r_\Delta = 2$) in $T < T_{N2}$ (Phase II), and (iii) a single-Q state ($r_\Delta = 0$) in $T_{N2} < T < T_{N1}$ (Phase III). The consistency with Fig. 2 strongly supports the continuous multi-Q manifold interpretation, linking single-Q and $C_{3z}$-symmetric triple-Q orderings with distinct chiral and nematic properties. Ground state visualizations are shown in Fig. 3h. The field-induced Phase II is stabilized by the larger residual magnetic moment of the three-fold symmetric triple-Q phase (see the magnetization data in Fig. 1d), favoring a state in which one of the four spin sublattices aligns along $H$. A residual net moment exists in both Phases II and I (even at zero field) due to the anisotropy terms in our model, and its direction ($\pm \hat{c}$) is linked to the sign of the underlying spin chirality.

We now discuss Phase IV and the limitations of our model. The absence of both MLD and MCD in Phase IV implies preserved $C_{3z}$ symmetry, suggesting a non-chiral triple-Q state. A likely configuration is the up-up-up-down (UUUD) structure, a collinear triple-Q ordering (Fig. 3d) observed in triangular lattice antiferromagnets under a magnetic field[41]. In classical simulations, the UUUD phase requires a magnetization of half the saturation value ($M = 1.5~\mu_B/Co^{2+}$), thereby appearing at higher fields in our model compared to $H_m$ ~3.5 T (Phase IV exhibits $M = 0.2~\mu_B/Co^{2+}$ at 7 T). We attribute this discrepancy to longitudinal spin fluctuations not captured in our classical model.

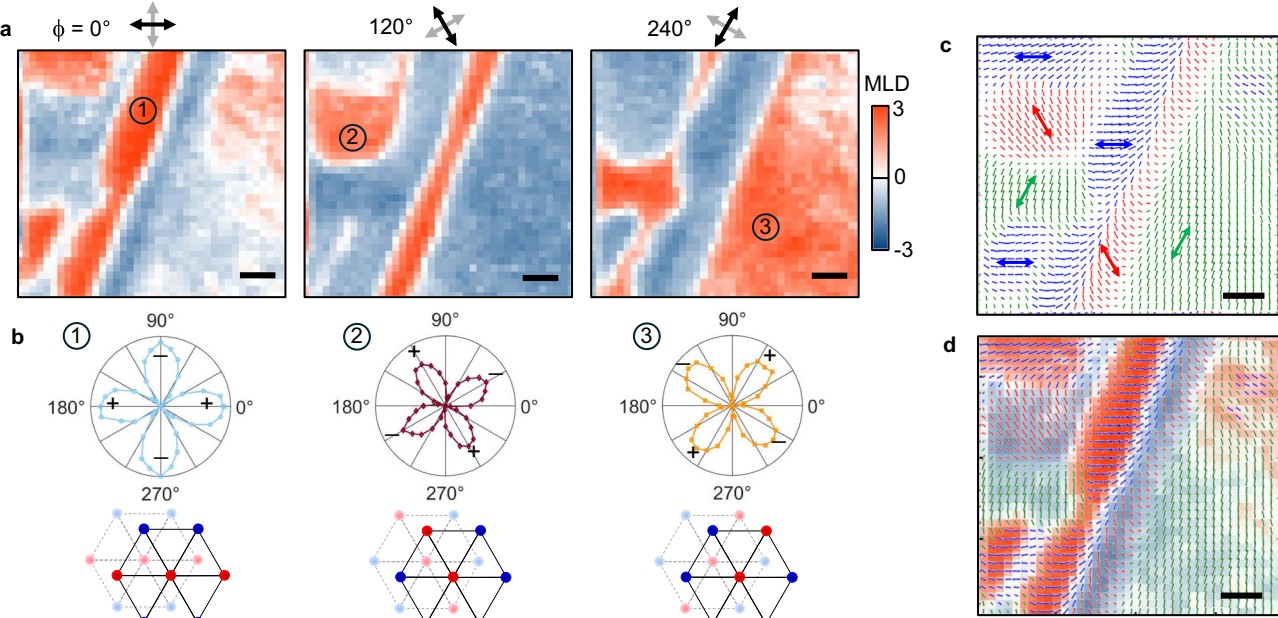

**Fig. 4 | Spatially-resolved images of nematic (single-Q) AFM domains in $Co_{1/3}TaS_2$, acquired via scanning MLD microscopy. a** Three $76 \times 70\ \mu m$ images of MLD at $T \approx 27$ K and $H=0$ and (in phase III), using linearly-polarized light modulated between $\phi = 0°/90°$, $120°/210°$, and $240°/330°$ with respect to the $Co_{1/3}TaS_2$ $a$-axis. Depending on $\phi$, different spatial regions show large and positive MLD signal. (Note: from each image, an MLD image in the nonmagnetic phase at 38 K was subtracted, reducing background signals). All scale bars are 10 $\mu m$. **b** Polar plots of

the measured MLD versus $\phi$, measured at the three locations indicated. These polar plots demonstrate $C_{2z}$ symmetry of the single-**Q** (stripe) AFM order, mutually rotated by 120°, and depicted in the diagrams below where red/blue dots indicate Co spins oriented into/out of the page. **c** Spatial map of the measured nematic director, revealing domains oriented along the indicated directions. **d** Nematic director map overlaid with the first MLD image shown in **a**.

Indeed, the ordered magnetic moment of 1.3 $\mu_B/Co^{2+}$ observed at 3 K–much smaller than the classical 3 $\mu_B/Co^{2+}$–indicates substantial spin fluctuations[19]. A possible consequence of strong fluctuations in triangular lattice antiferromagnets is a site-dependent renormalization of ordered moments[42], e.g., the three "up" moments could become shorter than the single "down" moment (Fig. 3e), reducing net magnetization. Precise determination of Phase IV requires future neutron diffraction measurements under magnetic field.

**Imaging nematic and chiral antiferromagnetic domains**

A notable benefit of optical methods is the ability to directly image real-space profiles of different magnetic states and their domains. In Figs. 4 and 5, we employ MLD and MCD microscopy (see Methods) to spatially resolve, respectively, the discrete $Z_3$ nematic domains arising from single-**Q** AFM order, and the binary domains associated with positive and negative spin chirality. Figure 4a shows three MLD images of the same area, acquired in Phase III (single-**Q**), using probe light linearly polarized at $\phi = 0°$, 120°, and 240° with respect to the crystal $a$-axis. The images reveal distinct regions with strong positive MLD (red) and weaker negative MLD (light blue). Importantly, the red regions are *different* in each image. Largest MLD occurs when $\phi$ aligns parallel to the stripes of single-**Q** magnetic order. The images therefore indicate three different single-**Q** domains where the stripes align along 0°, 120°, and 240°, as expected from a triangular lattice. The $C_{2z}$ symmetry of the domains is demonstrated in Fig. 4b, where MLD at the three indicated locations is measured versus $\phi$. Each polar plot exhibits $C_{2z}$ in-plane anisotropy, oriented approximately along 0°, 120°, and 240° (see also Supplementary Fig. S4). These images confirm $Z_3$ nematic order in $Co_{1/3}TaS_2$, induced by single-**Q** AFM order, with different nematic domains having ordering wave vectors $\mathbf{Q}_v$ related by $\pm 120°$ rotation.

Analyzing the MLD at each position yields a spatial map of the nematic director (Fig. 4c). The directors indicate the angle $\phi$ giving maximum MLD signal, and have lengths proportional to the MLD

magnitude. For clarity, blue, red, and green directors represent orientations closest to 0°, 120°, and 240°. This classification scheme remains consistent at different locations on the sample (see Supplementary Fig. S5). Notably, the nematic domains can be quite large, extending nearly 1 mm. Figure 4d shows the director map overlaid on the MLD map of Fig. 4a.

Interestingly, repeated thermal cycling into the paramagnetic state, even staying at 300 K for days or weeks, did not affect the nematic domain patterns. It suggests that they are pinned by extrinsic factors that locally break the underlying hexagonal crystal symmetry, such as local strain. Similarly, $Fe_{1/3}NbS_2$ and $FePSe_3$ have also shown biasing of an underlying three-state AFM nematicity by uniaxial strain[22,24]. Furthermore, the MLD signals remain essentially unchanged below $T_{N2}$ where spin chirality also emerges (*cf.* Fig. 1g), suggesting a smooth transition between the nematic order in Phase III and Phase I. This implies that at $T_{N2}$, single-**Q** domains with $\Delta_i \neq 0$ and $\Delta_{j,k} = 0$ ($i$, $j$, $k \in \{1, 2, 3\}$) transform continuously into distorted triple-**Q** domains with $\Delta_i > \Delta_j = \Delta_k > 0$.

Finally, we use MCD microscopy to study spontaneous formation of chiral AFM domains (Fig. 5). MCD images in the paramagnetic state (40 K, Fig. 5a) show no signal, as expected. However, Fig. 5b and d show that $Co_{1/3}TaS_2$ can be completely poled to a positive or negative chiral state by cooling into Phase I in applied $H$ as small as $\pm70$ mT. Most importantly, Fig. 5c shows that small and irregularly-shaped chiral domains spontaneously form when cooled in $H = 0$. Their characteristic size is much smaller than the nematic domains. Also in marked contrast to the nematic domains, the chiral domain pattern changes randomly after thermally cycling above $T_{N1}$, suggesting that the underlying nematicity of the distorted triple-**Q** order does not bias the handedness of spontaneously-forming chiral domains. Figure 5e shows chiral domains at a different location, again showing a random pattern. These images confirm that, to within our microscope's 1 micron spatial resolution, Phase I is not a phase-separated mixture of

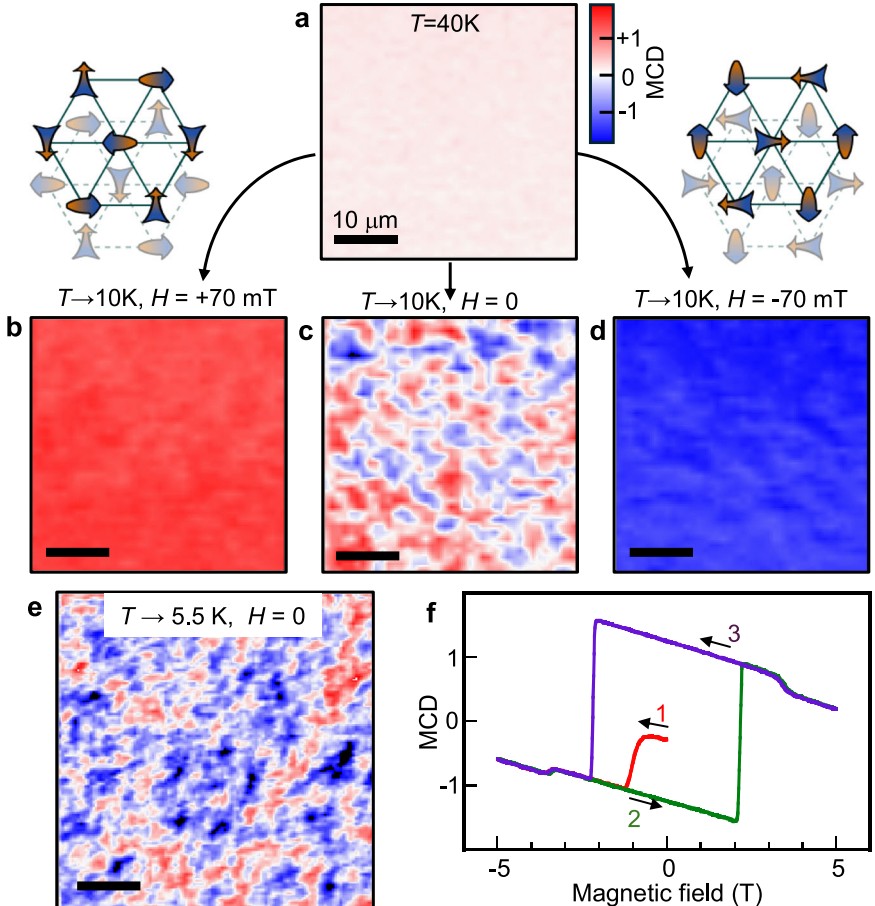

**Fig. 5 | Spatially resolving spontaneously-forming chiral AFM domains in Co$_{1/3}$TaS$_2$, via MCD microscopy. a** 40 × 40 μm MCD image at $T$ = 40 K (in the paramagnetic state) shows no signal. **b–d** Same, but after cooling slowly to $T$ = 10 K (Phase I) in $H$ = +70 mT, 0 mT, and −70 mT. While cooling below $T_{N2}$ in small $H$ readily poles the sample to uniform positive or negative chirality, cooling in zero field results in a dense pattern of spontaneously-formed chiral domains. Scale bars are 10 μm. The diagrams depict two distorted triple-**Q** spin configurations related by time-reversal, which have opposite scalar spin chirality but the same nematicity. Small/large arrowheads are canted into/out of the plane. **e** A 50 × 50 μm MCD image taken at a different location on the sample, following a more rapid zero-field cooldown to 5.5 K. **f** MCD($H$) scan after initially cooling the sample to 15 K in $H$ ≈ 0 (i.e., starting from a configuration with chiral domains). The probe beam is large (1 mm). $H$ is ramped from 0 → − 5 → + 5 → − 5 T. The initial magnetization curve saturates quickly, at a field much smaller than the chiral switching field (≈2 T) of the main hysteresis loop.

coexisting single-**Q** and triple-**Q** ground states. Finally, Fig. 5f shows that, starting from a random configuration, the chiral domains are readily poled by fields much less than the coercive field required to switch a fully polarized sample, suggesting weak pinning.

## Discussion

By combining magneto-optical techniques with theoretical analysis, we have elucidated the coexisting chiral and nematic properties in the multi-**Q** antiferromagnet Co$_{1/3}$TaS$_2$. Through temperature- and field-dependent MCD and MLD measurements, we identified the four distinct antiferromagnetic phases, each defined by the presence or absence of spin chirality and nematicity. Thermal fluctuations suppress chirality above $T_{N2}$, favoring collinear magnetic order, while magnetic fields restore the $C_{3z}$ rotational symmetry and thus suppress nematicity. These observations are captured by a minimal spin model incorporating four-spin interactions and magnetic anisotropy, which gives rise to a continuous multi-**Q** manifold. Real-space imaging using MCD and MLD microscopy revealed robustly-pinned domains with $Z_3$ nematic directors, alongside much smaller, irregularly shaped chiral domains that are easily reoriented by modest magnetic fields—suggesting a weaker pinning mechanism. These contrasting domain behaviors highlight the distinct natures of nematic and chiral order in Co$_{1/3}$TaS$_2$. Overall, this work demonstrates the efficacy of magneto-

optical techniques for characterizing symmetry and chirality in complex spin textures, suggesting applications to systems where transport-based methods are less applicable (e.g., insulators). The ability to detect and image both nematicity and spin chirality offers a powerful avenue for investigating multi-**Q** magnetism, with implications for understanding and engineering topological magnetic states.

## Methods

### Sample synthesis and characterization

Single-crystal Co$_{1/3}$TaS$_2$ was synthesized using a standard chemical vapor transport method applied to polycrystalline Co$_{1/3}$TaS$_2$, as detailed in refs. 19,25,43. Since the magnetic properties of Co$_{1/3}$TaS$_2$ are sensitive to Co compositions[43], careful assessments of the Co composition were conducted. The $T_{N2}$ value, which reaches a maximum of 26.5 K with a minimized extent of Co vacancies[43], serves as a reliable indicator of sample quality. Only samples exhibiting $T_{N2}$ = 26.5 K were used for this study, with an estimated vacancy concentration of less than 3%.

### MCD and MLD measurements

The experimental setup, depicted in Fig. 1b, used wavelength-tunable probe light (typically 650 nm or 700 nm) derived from a white light source (xenon lamp) spectrally filtered through a 300 mm

spectrometer. The probe light was mechanically chopped at 137 Hz, linearly polarized, and then polarization-modulated by a photoelastic modulator (PEM). For MCD, the polarization was modulated between right- and left-circular (±quarter-wave modulation) at 50 kHz. For MLD, the polarization was modulated between linear and cross-linear (±half-wave modulation) at 100 kHz. The $Co_{1/3}TaS_2$ samples were mounted in helium vapor in the variable-temperature (2–300 K) insert of a 7 T split-coil magnet with direct optical access. The probe light was weakly focused on the sample (≈1 mm spot size) at near-normal incidence, and the reflected light intensity was measured by an avalanche photo-diode, amplified, and demodulated using two lock-in amplifiers. We confirmed that perfectly normal incidence ($H \parallel \mathbf{k} \parallel c$) gave the same results. The MCD experiment measured the normalized difference between right- and left-circularly polarized reflected intensities, $(I_R - I_L)/(I_R + I_L)$. Similarly, MLD measured $(I_\phi - I_{\phi+90°})/(I_\phi + I_{\phi+90°})$, where $\phi$ is the angle of the probe's linear polarization. We note that MCD is a close relative of the magneto-optical Kerr effect (MOKE), and in particular the Kerr ellipticity, and is sensitive to magnetic order(s) that generate non-zero off-diagonal conductivity $\sigma_{xy}(\omega)$. Complementing MCD, MLD is sensitive to in-plane anisotropy of the optical conductivity [e.g., $\sigma_{xx}(\omega) - \sigma_{yy}(\omega)$], which can arise from single-$\mathbf{Q}$ (stripe-like) AFM order.

For imaging experiments requiring high spatial resolution, the light source was a 650 nm superluminescent diode, and the samples were instead mounted on the vacuum cold finger of a small optical cryostat. A high numerical aperture (NA = 0.55) microscope objective was used to focus the probe light at normal incidence down to ≈1 μm spot, that could be raster-scanned across the sample surface. Small out-of-plane magnetic fields up to 200 mT were applied using external permanent NdFeB magnets.

## Classical Monte Carlo simulations

In addition to the zero-temperature magnetic phase diagram calculation (described in Supplementary Section 2E), the phase diagram as a function of temperature and out-of-plane magnetic field ($H$) was obtained using classical Monte Carlo simulations. The simulations employed a combination of the Langevin dynamics algorithm and simulated annealing. All calculations were conducted with the Sunny software package[44,45]. To achieve statistically well-averaged results, 20 replicas of $36 \times 36 \times 6$ sized $Co_{1/3}TaS_2$ supercell (15,552 Co sites) were prepared and simulated in parallel. The spin systems were initialized by field cooling from 50 K to ensure a uniform alignment of the scalar spin chirality sign, which was necessary to achieve consistent results due to the intertwined nature of the transition between Phases I and II and the chirality sign. At a given temperature and field, we sampled the time evolution of each spin system using the Langevin dynamics after 5000 Langevin steps for initial thermalization. The Langevin time step and damping constant were set to $\frac{0.05}{S^2(J_1 + 0.1H)}$ (meV$^{-1}$) and 0.1, respectively, where $S = 3/2$, $J_1 = 1.212$ meV[21], and $H$ is expressed in units of Tesla. From the collected samples, we calculated the magnitudes of the three Fourier components ($|\widetilde{\mathbf{S}}_{\mathbf{Q}_\nu}| \equiv \Delta_\nu$, see Eq. (1) or Supplementary Section 2A) via Fourier transformation. The results are presented in Fig. 3f, g.

## Data availability

Experimental Source data are provided with this paper.

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

## Acknowledgements

E.K. and S.A.C. gratefully acknowledge support from the Los Alamos LDRD program and the U.S. Department of Energy (DOE), Office of Science, Quantum Science Center. P.P. acknowledges support from the U.S. DOE, Office of Science, Basic Energy Sciences, Materials Science and Engineering Division. J.-G.P. acknowledges support from the Samsung Science & Technology Foundation (Grant No. SSTF-BA2101-05) and the Leading Researcher Program of the National Research Foundation of Korea (Grant No. 2020R1A3B2079375 and RS-2020NR049405). C.D.B. was supported by the U.S. DOE, Office of Science, Office of Basic Energy Sciences, under Award Number DE-SC0022311. The National High Magnetic Field Laboratory is supported by National Science Foundation (NSF) DMR-1644779, the State of Florida, and the U.S. DOE. E.K. wants to thank Z. Hawkhead for fruitful discussion.

## Author contributions

S.A.C., C.D.B., P.P., and J.-G. P. initiated the project. P.P. synthesized the single-crystal samples. S.A.C. and E.K. performed the optical experiments. C.D.B., P.P., and W.C. developed the theoretical model. S.A.C., P.P., C.D.B., and E.K. wrote the manuscript with contributions from all authors.

## Competing interests

The authors declare no competing interests.
