## [Transparent Peer Review file · Nature Communications]

Tunable chiral and nematic states in the triple-Q antiferromagnet $\text{Co}_{1/3}\text{TaS}_2$

Corresponding Author: Dr Scott Crooker

Version 0:

Reviewer comments:

Reviewer #1

(Remarks to the Author)

This study investigates the complex multi-Q magnetic textures of the triangular antiferromagnet $\text{Co}_{1/2}\text{TaS}_2$ using optical techniques, namely magnetic circular dichroism (MCD) and magnetic linear dichroism (MLD). By combining MCD and MLD, the authors are able to distinguish between chiral order and nematic order, thereby corroborating and complementing the results of neutron experiments. Moreover, owing to the excellent real-space resolution of MCD and MLD, the authors have resolved the real-space magnetic domain information and further determined the magnetic structure of this system. In addition, this technique has generality and is expected to be extended to other complex systems as one of the technical means for identifying intricate magnetic textures.

Overall, this is a thorough work and is worthy of publication. I have only a few minor comments to address:

1. In the Introduction, the authors state that antiferromagnets can exhibit a variety of complex spin structures, whereas ferromagnets cannot. This seems to involve some ambiguity in definitions. For example, MnSi, a famous skyrmion host, is believed to have a weak itinerant ferromagnetic ground state.
2. In Fig. 1(e), the MCD signal between TN2 and TN1 remains visible (albeit weaker compared to the signal below TN1). Does this imply the presence of a potential chiral magnetic structure in this temperature range or other possible effects? The authors should clarify this point.

Reviewer #2

(Remarks to the Author)

Erik Kirstein and coworkers have investigated the magnetic phases of the van der Waals material CoTa_3S_6 , using optical microscopy (MLD/MCD), transport measurements and magnetic simulations including Monte Carlo methods. The manuscript is well-written and the figures are clear. The new finding seems to be a mixed low T zero field phase, where 1Q and 3Q magnetic order coexists. The authors show laterally resolved optical data with a resolution of about 1 micron, where the MLD and MCD signals are used as a measure of 1Q and 3Q magnetic character, respectively.

A main weakness of the manuscript is that the authors do not properly cite and relate their work to the existing literature, making it unnecessarily difficult for the reader to discriminate possibly new results from already established ideas and facts. For instance, the concept of a continuum of states between 1Q, 2Q and 3Q, has already been put forward in Ref. 38, however, a comparison between the proposed two-dimensional "multi-q manifold" and the one-dimensional mixing scheme of Ref. 38 is not made. Moreover, part of the phase diagram (Fig. 2c) is based on results from a previous publication of the authors (Ref. 19) and results from the Tokyo group (Ref. 18).

The proposed distorted 3Q state in phase I -- predicted for hcp Mn/Re(0001) in Ref. 38 (without experimental proof) and there explained based on higher than 4th-order exchange interactions (beyond 4-spin or biquadratic exchange) is here argued to be the result of crystalline anisotropy without taking other possibilities into account. Moreover, I don't find the argument for such a state convincing: If phase I constitutes a 3Q state with some 1Q component, why is then the MLD signal (which is introduced by the authors to be a measure of 1Qness) at a maximum in phase I, higher than in the supposedly pure 1Q phase III?

Overall, the manuscript does not clearly offer substantially new results which are backed by experiments to warrant

publication in Nature Communications.

In addition, I would like to raise more specific criticism:

1. In contrast to "chirality", the authors use the term "nematicity" without properly defining it. For that reason, stating that the 1Q AFM order gives rise to nematic domains does not explain nor add anything.
2. What is the exact relationship between 1Q AFM order (row-wise AFM) and the MLD and MCD signal? What is the exact relationship between 3Q order (109° between all neighboring spins) and the MCD and MLD signal?
3. Which MLD and MCD signals are expected for the coplanar 120° Neel state proposed for CoTa₃S₆ in Ref. 25?
4. page 2, "coexistence of single-Q and triple-Q orderings within the same system – an uncommon feature in real materials.": Firstly, a distorted 3Q state has been proposed already in Ref. 38. Secondly, 1Q, 3Q and transient 2Q states co-exist in the Mn/Re(0001) system due to different stacking order and inside domain walls, respectively.
5. page 2, "effective tetrahedral network of Co": This sentence implies that the magnetic coupling between Co atoms is identical within and across layers, which is certainly not the case.
6. "Continuous multi-Q manifold model": The mixed magnetic states in Ref. 38 are constructed in a one-dimensional fashion on a path on which the standard Heisenberg exchange is constant. Is this also the case for the authors' "manifold"?
7. page 4, "successfully captures the single-Q to triple-Q transition at TN2": Is it essential which 4th-order exchange term (biquadratic, 3spin, 4spin) is used for the transition to occur? Is the driving mechanism for this transition entropy?
8. page 5, "field-induced equilateral triple-Q state": It is not straightforward to make a distorted 3Q state more symmetric by applying an external B field. There is an energy term which might do the trick, but it is not mentioned in the manuscript.
9. page 7, "can be completely poled to a positive or negative chiral state": Is this behavior reproduced by simulations? Which energy term is responsible for the coupling of chirality and external magnetic field?
10. J. Spethmann et al. try to determine the orientation of their 3Q state in hcp Mn/Re(0001) and the accompanied DFT theory predicts a spin orientation in the 1Q state along the parallel spin-rows, driven by the magnetic dipolar interaction and anisotropic symmetric exchange. Is it possible to tackle these questions also for CoTa₃S₆.
11. Compared to SP-STM, where these states are imaged directly with atomic resolution, MLD and MCD are indirect measures, averaging over large surface areas. The effect of domain sizes and chemical inhomogeneity with a lateral size below the resolution limit is not discussed in the manuscript.
12. What is the information depth of the used techniques? Can a scenario be ruled out where the magnetic state at the surface is different from the bulk?
13. What can be concluded for the magnetic coupling between the layers experimentally? Is the proposed 109° coupling in phase I an assumption or a conclusion?
14. Finally, can the proposed uuud state (phase IV) be reproduced in the simulation?

Reviewer #3

(Remarks to the Author)

The work by Kirstein et al. presents a comprehensive study of the chiral and nematic states in the antiferromagnet Co_{1/3}TaS₂. They mapped out the antiferromagnetic phase diagram under different magnetic fields and temperature ranges using MLD and MCD measurements. A continuous multi-Q manifold model is proposed to understand the phase diagram. Finally, they presented spatially resolved chiral and nematic domains using MCD and MLD microscopy.

I think this paper is well written, and the results are scientifically sound. This work not only demonstrates that Co_{1/3}TaS₂ hosts rich nematic and chiral multi-Q states, but also represents a high-quality study of antiferromagnetic textures using optical techniques. I therefore recommend its publication in Nature Communications. Before publication, I have a few comments for the authors to improve the presentation of their manuscript.

Comments:

1. Phase I exhibits both MCD and MLD signals. The authors propose that this phase corresponds to a non-equilateral triple-

Q state. Could the authors clarify how they rule out the possibility that Phase I arises from the coexistence of spatially separated single-Q and triple-Q domains? In such a case, both MCD and MLD signals could also be expected.

2. Do the authors have an intuitive physical picture or a simple explanation for why the magnetic textures in Phase I exhibit both nematicity and chirality? Although the authors attribute this to magnetic anisotropy, the explanation remains unclear. Magnetic anisotropy is relatively common, but the coexistence of nematic and chiral textures is much less so. What is special about $\text{Co}_{1/3}\text{TaS}_2$ in this context?

3. The MCD curve versus magnetic field in Fig. 2a shows a tilt, suggesting that the field induces some MCD even at high temperatures such as 30 K or 40 K. In contrast, at high temperatures (e.g., 30 K), the MCD is labeled to vanish at large fields in Fig. 2c. The authors should clarify this apparent inconsistency.

4. I have some questions regarding Phase IV: (i) In Section A, the authors state that magnetic fields induce two new phases, Phase II and Phase IV. However, this is not evident from Fig. 1. (ii) There is no visible sharp transition in Fig. 2b at high temperatures. How do the authors determine the boundary between Phase III and Phase IV in Fig. 2c?

5. Figures 4(c) and 4(d) appear repetitive. Moreover, there is no description of Fig. 4(d) in the main text. The authors should address this omission.

Version 1:

Reviewer comments:

Reviewer #1

(Remarks to the Author)
publish as is.

Reviewer #2

(Remarks to the Author)

Despite the substantial and detailed criticism I have raised in the first review round, Kirstein and coworkers have resubmitted a manuscript with only cosmetic changes. In their reply, most of my questions are evaded, some given answers are wrong and the clarifications and explanations which were indeed helpful to understand the work, did unfortunately not make it into the new version of the manuscript. I do understand the tactics of the rebuttal, but this way of dealing with justified critique is risky, and I remind the authors that two of them have already published a wrong conclusion about the magnetic state of the very same system in Ref. [25]. Therefore, most of my critique is still valid and I come to the same conclusion, that I do not see substantially new physics within the manuscript which is conclusively backed up by experiment to warrant publication in Nature Communication. In my opinion, certain inconsistencies of the manuscript need to be addressed before publication in any journal. Whether these problems can be fixed by simply writing more clearly or point toward deeper flaws of the work, I do not know.

For the future, I encourage the authors to refrain from the overuse of italic, bold and underlining in the review correspondence. Reviewers are quite capable of reading texts carefully without such "help".

In the following, I will not again comment on all problems of the manuscript and shortcomings of the rebuttal, but only address the ones I deem most critical.

1. Let us start with this claim from the rebuttal: "Moreover, we respectfully emphasize that the new findings include not only the experimental identification of a distorted triple-Q antiferromagnetic phase with co-existing chiral and C_{2z} symmetries (noted by the Reviewer), but also 1) the demonstration that magneto-optical methods for circular and linear dichroism can function as incisive reporters of these antiferromagnetic symmetries, ..."

At face value, this sentence is circular: Fundamentally, you cannot establish a new magnetic technique using a specific spin texture and then also claim that the observed spin texture is a new finding. Now, it seems that CoTa_3S_6 has four different magnetic phases: which spin textures are used to calibrate MLD and MCD and which results are actually new?

2. Another problem puts a main result of the work into question. Here, again a sentence from the rebuttal: "Finally, we note that the MLD signal (a reporter of 1Q order parameter) emerges and grows throughout Phase III, achieving a maximum at the Phase I – Phase III boundary, which remains approximately constant as the sample is cooled throughout Phase I (i.e., the MLD is not larger in Phase I as the Reviewer suggests). Figure 1g shows this reasonably well, and Supplemental Fig. S2d shows this extremely clearly."

This description is wrong! In both plots the highest MLD signal is found at the lowest temperature of $T=4.2$ K, not at $T=26.5$ K. But let us assume for the sake of argument, that the MLD signal is constant in phase I; even then the problem arises that the MLD signal interpreted as a direct measure of 1Q character is INCONSISTENT with the interpretation of phase III as a pure 1Q state and phase I as a mixed 1Q-3Q state. When the state becomes more 3Q-like in phase I, the 1Q character needs to drop. There seems to be a further temperature dependence which is not discussed and not accounted for.

3. As an answer to my comment #4 [page 2, "coexistence of single-Q and triple-Q orderings within the same system – an uncommon feature in real materials.": Firstly, a distorted 3Q state has been proposed already in Ref. 38. Secondly, 1Q, 3Q and transient 2Q states co-exist in the Mn/Re(0001) system due to different stacking order and inside domain walls, respectively] the authors write:

"As the authors of [38] will surely agree, such an unusual distorted 3Q state is indeed not common – we carefully searched through the existing literature for relevant precedent, and found that Mn/Re was one of only a very few examples containing related physics."

This is a very misleading statement, and the term "as the authors will surely agree" is pure speculation. There are only a handful of papers out there dealing with 3Q states at all (in the sense of nearest neighbor spins having angles close to 109.5° , not in the sense of skyrmion lattices and such). As far as I know, the first experimental observation was in the Mn/Re(0001) system by SP-STM, where 1Q and 3Q areas co-exist (due to different possible stacking order of the Mn layer) and in addition the 3Q state itself might be distorted (Haldar et al.). The former work is not even cited, although it is quite relevant for the discussion.

One cannot say whether something is common or uncommon if there are only a handful of cases in total. Furthermore, one might say that the ideal 109.5° -3Q state will never be observed in real systems, because there are always tiny interactions present which distort it.

4. The limited spatial resolution of MLD and MCD is an issue which is unfortunately not openly discussed in the manuscript. For a single image data point, the information seems to come from a volume 50-100 nm in depth and about 1000 nm in diameter. I agree with referee #3 that the authors do not convincingly exclude a scenario for phase I where separate 1Q and 3Q domains co-exist beyond the resolution limit, compared to the proposed laterally homogenous and itself distorted 3Q state (with 1Q component).

5. Concerning the proposed states for phase I and II, the comments in the rebuttal were quite helpful. I try to be brief here, but I think the present manuscript is not clear enough at this point. The authors need to clearly discriminate at least these three states: 1) the ideal, fully symmetric 109.5° -3Q state, 2) the state with one spin up and three spins symmetrically distorted, sometimes called the "threefold symmetric 3Q" and 3) the 3Q state which is distorted by crystal anisotropy, which looks like a 2Q state seen from the top. There might be more. In any case, I was confused, because the authors mark the central (red) point on the manifold in Fig. 3a as the three-fold symmetric 3Q. But at this point on the manifold is the ideal 109.5° -3Q. As far as I understand, the state proposed for phase II is not even on the manifold, because it has a higher Heisenberg energy than the ideal 109.5° -3Q.

Two more comments: starting from the ideal 109.5° -3Q state, uniaxial anisotropy alone does not produce a net moment. Secondly, the authors might want to consider the anisotropic symmetric exchange interaction, which can effectively couple the 3Q state in the way they propose for phase II (1 spin up), see F. Nickel et al., new Ref. [51].

6. Finally, as the authors are aware, higher-order interactions (HOI) such as the bi-quadratic one are needed to get a non-coplanar state such as the 3Q state, because an interaction is needed which favors 90° angles. I have not found any reference or mention of HOIs in the main text of the manuscript. This is odd, because without HOI neither phase I and phase II, nor the T-driven transition from phase I to phase III can be understood.

Reviewer #3

(Remarks to the Author)

The authors have satisfactorily addressed all my comments. I do recommend the manuscript for publication in Nature Communications.

Reviewer #4

(Remarks to the Author)

The revised manuscript is much clearer and addresses the points raised in the first round.

For Reviewer 1's concerns, the phrasing in the Introduction has been adjusted and the small MCD signal between TN2 and TN1 is now properly explained as cross-talk.

For Reviewer 2's main objections about Phase I, the authors have clarified why higher-order exchange interactions cannot explain the data, and why an anisotropy-based model is more consistent with neutron diffraction and magnon gap results. The clarification of the MLD signal behavior is also helpful. The improved definitions and terminology (nematicity, distorted vs three-fold symmetric 3Q) make the presentation stronger.

For Reviewer 3's questions, the authors have explained why Phase I is not phase coexistence, but a single distorted 3Q state with both chirality and nematicity. The additional discussion of anisotropy, the field-induced phases, and the clarifications in the figures all improve readability.

Overall, the work convincingly demonstrates how combining MCD and MLD can disentangle chiral and nematic orders and provide real-space imaging of magnetic textures. This is a strong contribution, and I recommend acceptance.

Version 2:

Reviewer comments:

Reviewer #2

(Remarks to the Author)

I thank the authors again for their detailed answers to my criticism. I think we had enough review rounds and the manuscript improved to some extent in the process. At this stage we should simply proceed according to the majority vote. My concerns stand and will be documented in the Peer Review File for future reference. I hope that the manuscript will age better than Ref. 25 and wish everyone a peaceful holiday.

Reviewer #3

(Remarks to the Author)

I think the authors have done a good job addressing the referees' comments. The additional suggestions and remarks from Referee 2 are also very helpful. If possible, it would be great for the authors to incorporate Referee 2's input into the manuscript.

Referee 2 also raises concern about the possible coexistence of 1Q and 3Q domains, particularly given the limited image area. If feasible, the authors could discuss the spatial resolution of their measurements and provide additional imaging data to further address this point.

Overall, I believe this work remains a high-quality study of antiferromagnetic textures using optical techniques. I continue to support the publication of this paper in Nature Communications.

Reviewer #4

(Remarks to the Author)

The authors have provided thoughtful and scientifically grounded clarifications to the previously raised points, particularly regarding the distinctions among the triple-Q states, the interpretation of the MLD signal, and the treatment of higher-order interactions.

In light of these revisions, I am satisfied that the manuscript has been strengthened, and I would be supportive of its publication.

Reply to Reviewers [manuscript NCOMMS-25-30582-T, “Tunable chiral and nematic states in the triple-Q antiferromagnet Co_{1/3}TaS₂”]

We are grateful to all three Reviewers for their careful reading of the manuscript, and we sincerely appreciate the positive assessments of our work from Reviewers #1 and #3 and for their recommendations to publish. The revised and resubmitted manuscript incorporates the suggested changes for improvement; see our point-by-point response to all Reviewer feedback below, and Changes to the Manuscript listed at the end. For ease of reading, all Reviewers’ original remarks are reprinted in *blue italics*, and changes to the manuscript appear in **red**.

Yours sincerely, *the authors*

Reply to Reviewer 1

This study investigates the complex multi-Q magnetic textures of the triangular antiferromagnet Co_{1/3}TaS₂ using optical techniques, namely magnetic circular dichroism (MCD) and magnetic linear dichroism (MLD). By combining MCD and MLD, the authors are able to distinguish between chiral order and nematic order, thereby corroborating and complementing the results of neutron experiments. Moreover, owing to the excellent real-space resolution of MCD and MLD, the authors have resolved the real-space magnetic domain information and further determined the magnetic structure of this system. In addition, this technique has generality and is expected to be extended to other complex systems as one of the technical means for identifying intricate magnetic textures. Overall, this is a thorough work and is worthy of publication. I have only a few minor comments to address:

Reply: We appreciate the Reviewer’s positive assessment of the manuscript and recommendation to publish (after addressing the Reviewer’s feedback in the revised manuscript), and especially for noting the general applicability of these techniques to other complex magnetic textures.

1. In the Introduction, the authors state that antiferromagnets can exhibit a variety of complex spin structures, whereas ferromagnets cannot. This seems to involve some ambiguity in definitions. For example, MnSi, a famous skyrmion host, is believed to have a weak itinerant ferromagnetic ground state.

Reply: We thank the Reviewer for pointing out that our phrasing in the Introduction could be interpreted as dismissive of non-trivial ferromagnetic orders. This was not our intent; rather we meant only that antiferromagnets can generally be described by one or more *finite-Q* orderings, whereas ferromagnets generally exhibit uniform net magnetization ($Q=0$). But as the Reviewer correctly notes, skyrmion phases in otherwise ferromagnetic materials are obviously not trivial. To avoid any misinterpretation, and without loss of generality, **the revised manuscript now simply omits the phrase “Unlike ferromagnets...” so that the third sentence now begins “Antiferromagnets can exhibit diverse spin configurations...”**

2. In Fig. 1(e), the MCD signal between TN2 and TN1 remains visible (albeit weaker compared to the signal below TN1). Does this imply the presence of a potential chiral magnetic structure in this temperature range or other possible effects? The authors should clarify this point.

Reply: The very small MCD signal that can be seen in Fig. 1e (between TN2 and TN1) arises from a small amount of cross-talk between MCD and MLD signals, and not from any chiral magnetic structure. This cross-talk was discussed in detail in Supplementary Section I.B. and in Supplemental Fig. S2 of the original manuscript (a more vivid example is shown in panel S2c). **The revised manuscript makes this connection more explicitly clear, at the end of section I B where this cross-talk effect is first noted and where Fig. S2 is first referenced, by adding “This cross-talk also generates the very small MCD signal between TN1 and TN2 in Fig. 1e (see Supplementary Fig. S2)”**

Reply to Reviewer 2

Erik Kirstein and coworkers have investigated the magnetic phases of the van der Waals material CoTa3S6, using optical microscopy (MLD/MCD), transport measurements and magnetic simulations including Monte Carlo methods. The manuscript is well-written and the figures are clear. The new finding seems to be a mixed low T zero field phase, where 1Q and 3Q magnetic order coexists. The authors show laterally resolved optical data with a resolution of about 1 micron, where the MLD and MCD signals are used as a measure of 1Q and 3Q magnetic character, respectively.

Reply: We thank the Reviewer for their thorough reading of the manuscript. We're glad to know that the Reviewer finds the manuscript well-written and the data clear.

Moreover, we respectfully emphasize that the new findings include not only the *experimental* identification of a distorted triple-Q antiferromagnetic phase with co-existing chiral and C_{2z} symmetries (noted by the Reviewer), but *also* 1) the demonstration that magneto-optical methods for circular and linear dichroism can function as incisive reporters of these antiferromagnetic symmetries, 2) experimental demonstration that a three-state (Z3) nematic order parameter is what distinguishes the various *B*- and *T*-dependent phases in this archetypal triangular-lattice antiferromagnet, and 3) the experimental demonstration that the chiral and Z3 nematic domains can be directly imaged via microscopy. Moreover, the data are supported by a consistent theoretical framework (multi-Q manifold) that incorporates magnetic anisotropy.

A main weakness of the manuscript is that the authors do not properly cite and relate their work to the existing literature, making it unnecessarily difficult for the reader to discriminate possibly new results from already established ideas and facts. For instance, the concept of a continuum of states between 1Q, 2Q and 3Q, has already been put forward in Ref. 38, however, a comparison between the proposed two-dimensional "multi-q manifold" and the one-dimensional mixing scheme of Ref. 38 is not made.

Reply: We would like to emphasize that the primary focus and achievement of this work lies in the experimental characterization of multiple antiferromagnetic phases in $\text{Co}_{1/3}\text{TaS}_2$, and not in the development of a comprehensive theoretical model to explain every nuance of these new data. We did of course carefully study the theoretical model of Ref. [38] (by Haldar *et al*), and we recognized its importance and relevance to our work – *which is precisely why Ref. [38] was already cited in our original manuscript*. However, we understand the Reviewer's point of view, and agree that it could be more appropriate to cite this work sooner and more prominently. **Our revised manuscript now also cites Haldar *et al* in the Introduction section, as an example of the most relevant prior study (to our knowledge) of such evolving and tunable multi-Q antiferromagnetic orders. Moreover, we have expanded the discussion in Section II C and also in Section II D to let readers know how the theory of Haldar *et al* relates to our experimental findings (see Changes to Manuscript below).**

That said, *we stress that the main message of our manuscript is based on experimental discoveries*. For completeness our manuscript includes a new theoretical model that successfully captures our data, including the importance of magnetic anisotropy. But it is outside the scope of our experimental paper to compare and contrast the details of our model to the details of other theoretical models (such as the 6-spin interactions considered by Haldar *et al*). Others may do this if they wish, and indeed we hope that our results will motivate more theoretical work along these lines. We hope that the Reviewer understands and appreciates our rationale and approach.

To reiterate: Through the combined use of two magneto-optical probes (MLD and MCD), supported by previous characterization results, we were able to determine the chiral and, crucially, nematic magnetic symmetries of these distinct phases. The identification of both the three-fold symmetric and distorted

triple-**Q** states—based solely on experimentally inferred magnetic symmetry and spin chirality—is a key result of this study. These findings stand independently of which theoretical modelling we adopted and represent a significant new advance in the experimental investigation of complex spin textures.

Moreover, part of the phase diagram (Fig. 2c) is based on results from a previous publication of the authors (Ref. 19) and results from the Tokyo group (Ref. 18).

Reply: Yes, and as the manuscript explicitly states, the general structure of the magnetic phase diagram of Fig. 2c is already suggested by the recent magnetization, neutron, electrical transport, and susceptibility measurements of Refs. [18, 19, 25] (indeed, the manuscript clearly states that data shown in Figs. 1c,d is adapted from [19]). *We reiterate that the primary goal and main message of our manuscript is to experimentally elucidate the underlying nature and magnetic symmetries of these different magnetic phases, using optical methods based on circular and linear dichroism, which allows to classify the various phases by the presence/absence of chirality and C_{2z} symmetry (stripe-like order).*

The proposed distorted 3Q state in phase I -- predicted for hcp Mn/Re(0001) in Ref. 38 (without experimental proof) and there explained based on higher than 4th-order exchange interactions (beyond 4-spin or biquadratic exchange) is here argued to be the result of crystalline anisotropy without taking other possibilities into account. Moreover, I don't find the argument for such a state convincing: If phase I constitutes a 3Q state with some 1Q component, why is then the MLD signal (which is introduced by the authors to be a measure of 1Qness) at a maximum in phase I, higher than in the supposedly pure 1Q phase III?

Reply: Again we emphasize that we had carefully read Ref. 38 (Haldar et al) and recognized its relevance to the present work—indeed, *this is why it was explicitly cited in the relevant sections of the main text.* Regarding the sixth-order exchange interaction proposed in Ref. 38, we did attempt to incorporate such terms into our momentum-space modeling framework. However, this higher-order term failed to capture the experimental features of $\text{Co}_{1/3}\text{TaS}_2$. Specifically: 1) Increasing its strength resulted only in a first-order transition between the three-fold symmetric 3**Q** and 2**Q** states, without stabilizing an intermediate distorted 3**Q** state as a distinct ground state, and 2) While the observation above might still suggest room for a continuous manifold between 3**Q** and 2**Q** states (that is, $\Delta_i < \Delta_j = \Delta_k$) around the transition, our experimental observations instead point to a manifold connecting 1**Q** and 3**Q** phases (that is, $\Delta_i > \Delta_j = \Delta_k$) in $\text{Co}_{1/3}\text{TaS}_2$, an opposite side to this. For these reasons, we concluded that the sixth-order interaction proposed in Ref. 38 is unlikely to be responsible for our observations in $\text{Co}_{1/3}\text{TaS}_2$.

Our theoretical model instead attributes this phase to the very common (and thus more plausible) magnetic anisotropy, guided by a broad set of experimental constraints. Notably, neutron diffraction has identified an out-of-plane spin orientation in the single-**Q** phase and specific high-symmetry directions in Phase I. Additionally, the observed finite magnon gap in Phase I indicates full breaking of spin-rotational symmetry. The anisotropy terms included in our model—namely, the single-ion anisotropy (A) and the bond-dependent exchange anisotropy ($J_{\pm\pm}$) in Eq. (3) of the main text—were chosen carefully to cover all these findings. We would like to emphasize that, unlike four-spin or six-spin interaction terms, these second-order anisotropy terms are common in many magnetic materials.

Given (i) the inconsistency of the Ref. 38 model with our experimental results and (ii) the ability of our minimal spin model, based on well-established and commonly found anisotropy terms, to quantitatively capture the key features of $\text{Co}_{1/3}\text{TaS}_2$, we believe it offers a natural and sufficient explanation without the need to invoke higher-order interactions beyond four-spin terms.

Finally, we note that the MLD signal (a reporter of 1**Q** order parameter) emerges and grows throughout Phase III, *achieving a maximum at the Phase I – Phase III boundary, which remains approximately constant*

as the sample is cooled throughout Phase I (i.e., the MLD is not larger in Phase I as the Reviewer suggests). Figure 1g shows this reasonably well, and Supplemental Fig. S2d shows this extremely clearly.

Overall, the manuscript does not clearly offer substantially new results which are backed by experiments to warrant publication in Nature Communications.

Reply: Again, we respectfully remind the Reviewer that our manuscript describes new experimental results (Figs. 1, 2, 4, & 5), which are backed by theory -- not the other way around as the Reviewer's comment implies. Moreover, the theoretical framework that we developed fully captures the experiments, by incorporating common magnetic phenomena (such as anisotropy).

In addition, I would like to raise more specific criticism:

1. In contrast to "chirality", the authors use the term "nematicity" without properly defining it. For that reason, stating that the 1Q AFM order gives rise to nematic domains does not explain nor add anything.

Reply: By “nematicity” we assume the usual definition: an order parameter that breaks rotational symmetry. Nematicity in crystalline solids (as opposed to, say, liquid crystals), is technically a ‘discrete nematicity’, as it does not break a continuous rotational symmetry but rather a discrete lattice rotational symmetry. For a 2D triangular lattice, “nematic” therefore means a broken discrete rotational symmetry from C_6 to C_2 . In our manuscript we followed recent literature on single-Q (stripe-like) antiferromagnetism in hexagonal magnets [Nature Physics **20**, 1888 (2024); Nature Materials **19**, 1062 (2020)], and explicitly refer to the single-Q stripe-like AFM order in CoTaS as “discrete three-state” or “Z3” nematic order, since there are three possible choices of direction (ie, along the three M-ordering wavevectors). This definition is already explicitly stated in the manuscript, for example in the abstract and in the 2nd paragraph of the Introduction section. **Nonetheless, for added clarity we have added phrasing throughout the paper to emphasize the connection between C_{2z} symmetry, stripe-like single-Q antiferromagnetic order, and three-state (Z3) nematicity.**

We note that the concept of nematic order in solid crystals has a long history, for example nematic order in iron-based superconductors, where C_4 symmetry becomes C_2 (called “Ising-like” or “Z2” nematic order); see for example [Fernandes et al., Nature Physics **10**, 97 (2014)].

2. What is the exact relationship between 1Q AFM order (row-wise AFM) and the MLD and MCD signal? What is the exact relationship between 3Q order (109° between all neighboring spins) and the MCD and MLD signal?

Reply: We thank the Reviewer for this question, which has alerted us to a potential point of misunderstanding. We do **not** mean to imply that the tetrahedral 3Q order has perfect 109-degree angles between neighboring spins (the manuscript never said this). Firstly, the tetrahedral lattice defined by the Co atoms is itself not geometrically perfect (there are different inter- and intra-plane Co-Co distances), and moreover the intra- and inter-plane exchange couplings J_I and J_{cI} are different. Thanks to the Reviewer, we recognize that the words “equilateral” and “non-equilateral” might incorrectly suggest the presence/absence of geometrically perfect tetrahedra and perfect 109 degree spin order. Rather, what we mean is that our data reveal phases that are “three-fold symmetric triple-Q” (defined in our manuscript as $\Delta_1 = \Delta_2 = \Delta_3$, with chirality but without nematicity, such as Phase II), and phases that are “distorted triple-Q” (where $\Delta_{1,2,3}$ are not equal; these exhibit both chirality *and* nematicity such as Phase I). **To avoid any misinterpretation, we have updated our terminology, replacing “equilateral 3Q” with “three-fold symmetric 3Q”, and “nonequilateral 3Q” with “distorted 3Q”. Moreover, Fig. 1 and its caption now explicitly show/mention the different intra- and inter-plane exchange couplings J_I and J_{cI} .**

To answer the Reviewer's question: The single-Q "stripe" AFM order discussed in the manuscript (where Q is half a reciprocal lattice vector, or at the M-points of the hexagonal Brillouin zone) generates non-zero MLD because it breaks the rotational symmetry of the crystal (reduction to C_{2z} symmetry). Related phenomena were recently observed in hexagonal antiferromagnets magnets [e.g., Nature Physics **20**, 1888 (2024); Nature Materials **19**, 1062 (2020); Nano Letters **21**, 6938 (2021)] and arises from the resulting in-plane anisotropy of the optical conductivity. Such purely single-Q AFM order does not possess chirality, does not generate any Hall conductivity σ_{xy} , and therefore does not generate MCD.

In contrast, "three-fold symmetric 3Q order" (defined in our manuscript as $\Delta_1 = \Delta_2 = \Delta_3$) does possess chirality and therefore does generate σ_{xy} and MCD, but does not generate MLD because it does not break the discrete rotational symmetry of the crystal (i.e., does not possess C_{2z} symmetry). These concepts and definitions are described and clearly defined in the manuscript.

3. Which MLD and MCD signals are expected for the coplanar 120° Neel state proposed for CoTa3S6 in Ref. 25?

Reply: The coplanar Neel state proposed in Ref. [25] would be expected to give MCD (since σ_{xy} is allowed), but would not be expected to give MLD (since its symmetry is not C_{2z}).

4. page 2, "coexistence of single-Q and triple-Q orderings within the same system – an uncommon feature in real materials.": Firstly, a distorted 3Q state has been proposed already in Ref. 38. Secondly, 1Q, 3Q and transient 2Q states co-exist in the Mn/Re(0001) system due to different stacking order and inside domain walls, respectively.

Reply: Please see the discussion above relating to Ref. [38] by Haldar et al., **which is now cited earlier and more prominently in the revised manuscript**. Again, we stress that our manuscript already cited Ref. [38] -- now Ref. [26] in the revised manuscript -- precisely *because* of its relevance to this distorted triple-Q state. As the authors of [38] will surely agree, such an unusual distorted 3Q state is indeed not common – we carefully searched through the existing literature for relevant precedent, and found that Mn/Re was one of only a very few examples containing related physics.

5. page 2, "effective tetrahedral network of Co": This sentence implies that the magnetic coupling between Co atoms is identical within and across layers, which is certainly not the case.

Reply: We did not assume identical intra- and inter-layer magnetic couplings (as the theory section already made clear, via use of different couplings J_{\parallel} and J_{\perp}). Nonetheless, as discussed above in our reply to Comment #2, to make this even more explicitly clear and avoid any confusion, **Figure 1a is now updated with the intra-layer and inter-layer magnetic couplings (J_{\parallel} and J_{\perp}) now labeled**.

6. "Continuous multi-Q manifold model": The mixed magnetic states in Ref. 38 are constructed in a one-dimensional fashion on a path on which the standard Heisenberg exchange is constant. Is this also the case for the authors' "manifold"?

Reply: Thank you for the question. We consider the full two-dimensional (not one-dimensional) degenerate ground state manifold of the Heisenberg model (ie, constant Heisenberg exchange). This manifold obviously connects *all possible* single-, double-, and triple-Q states and the degeneracy persists regardless of the specific values of the Heisenberg exchange interactions, provided that their Fourier-transform, $J(\mathbf{q})$, is minimized at the M-point—consistent with the ordering wave vector observed in $\text{Co}_{1/3}\text{TaS}_2$.

7. page 4, "successfully captures the single-Q to triple-Q transition at TN2": Is it essential which 4th-order exchange term (biquadratic, 3spin, 4spin) is used for the transition to occur? Is the driving mechanism for this transition entropy?

Reply: It is not essential. As long as the chosen fourth-order interaction favors the triple- \mathbf{Q} ground state over the single- \mathbf{Q} state, its specific form does not affect the thermally-induced transition to single- \mathbf{Q} . As the Reviewer correctly notes, this transition is entropy-driven: thermal fluctuations favor collinear (single- \mathbf{Q}) states over non-collinear ones [Villain, J. *et al.*, Journal de Physique **41**, 1263 (1980)], thereby generating the transition regardless of the detailed form of the real-space four-spin interaction.

Our work does not attempt to identify the specific microscopic real-space four-spin interactions, which is a complex task for metallic materials like $\text{Co}_{1/3}\text{TaS}_2$. Instead, we formulate the problem in momentum space and retain only the uniform ($\mathbf{Q}_0=0$) and ordering wave vector components ($\mathbf{Q}_1, \mathbf{Q}_2, \mathbf{Q}_3$) of the spin field (see Supp. Note II. B). A given four-spin interaction in momentum space can arise from multiple distinct combinations of four-spin terms in real space. As a result, our analysis does not distinguish between biquadratic and more general four-spin interactions in real space. The simple scalar biquadratic form is chosen only for Monte Carlo simulations that require a concrete real-space model that reproduces the momentum space Hamiltonian restricted to the above-mentioned wave vectors. This choice of real space interactions is not relevant for the long wavelength static and low-energy properties of M-ordered systems. That said, it can affect the nature of field and temperature induced phases, such as phase IV.

8. page 5, "field-induced equilateral triple-Q state": It is not straightforward to make a distorted 3Q state more symmetric by applying an external B field. There is an energy term which might do the trick, but it is not mentioned in the manuscript.

9. page 7, "can be completely poled to a positive or negative chiral state": Is this behavior reproduced by simulations? Which energy term is responsible for the coupling of chirality and external magnetic field?

Reply: We thank the Reviewer for raising these questions. Since they are related, we provide here a single Reply that addresses both. The key mechanism behind the field-induced three-fold symmetric triple- \mathbf{Q} state (Phase II) is its higher magnetic susceptibility. This is strongly supported by the $M(H)$ magnetization curve of $\text{Co}_{1/3}\text{TaS}_2$ (see Fig. 1d), which shows an immediate gain of additional magnetization right after passing the metamagnetic transition field ($H > H_m$). Applying a large magnetic field along the c-axis restores the C_{3z} symmetry by stabilizing a state in which one of the four spin sublattices aligns with the field (as depicted in Fig. 3); this state has larger net residual moment.

Yes, the behavior is reproduced by our simulations (see Figs. 3f and 3g, for example). In our spin model, the coupling arises from the weak, residual net magnetic moment (present even at zero field; see Fig. 1d), which couples the external magnetic field to the sign of the scalar spin chirality. Such a moment in complex antiferromagnetic orders originates from either slight spin canting (due to magnetic anisotropy – **specifically, to the A and $J_{\pm\pm}$ anisotropy terms in our Equation 3**), or to orbital magnetization, and is commonly found in nontrivial antiferromagnetic spin orders that exhibit a spontaneous Hall conductivity (e.g., Mn_3Sn). The direction of the net moment is linked to the sign of the scalar spin chirality: domains with positive (negative) scalar spin chirality have net magnetization along $+\mathbf{z}$ ($-\mathbf{z}$). Coupling of this residual moment, and therefore chirality, to H is via Zeeman terms (see Eq. S20). This weak net moment and the resulting field-tunability of complex order parameters has been widely discussed in prior studies of antiferromagnets with spontaneous Hall effects: for instance, see Šmejkal *et al.* Nature Reviews Materials **7**, 482–496 (2022) and Chen *et al.*, Phys. Rev. B **101**, 104418 (2020).

We agree with the Reviewer that this coupling is an important aspect of the model and underlying physics, and the revised manuscript now more completely describes this coupling mechanism and the role of the anisotropy terms, in the second-to-last paragraph of Section II D.

10. J. Spethmann et al. try to determine the orientation of their 3Q state in hcp Mn/Re(0001) and the accompanied DFT theory predicts a spin orientation in the 1Q state along the parallel spin-rows, driven

by the magnetic dipolar interaction and anisotropic symmetric exchange. Is it possible to tackle these questions also for CoTa₃S₆.

Reply: We emphasize that the magnetic structure of bulk Co_{1/3}TaS₂ has already been determined (or at least strongly constrained) by neutron diffraction—the most powerful technique to determine/constrain microscopic spin orientations (Refs. 18–19). Our theoretical model was constructed to be fully consistent with these experimental findings. In particular, the magnetic anisotropy terms used in our spin Hamiltonian were chosen to reflect the experimentally observed spin orientations—such as the out-of-plane alignment of moments in the single-**Q** phase. These orientation profiles predicted by our model for each phase are explicitly shown in Figs. 3b–e. Indeed, spin orientations are constrained to certain high-symmetry directions and agree with the information imposed by the diffraction results from prior studies (Refs. 18–19).

11. Compared to SP-STM, where these states are imaged directly with atomic resolution, MLD and MCD are indirect measures, averaging over large surface areas. The effect of domain sizes and chemical inhomogeneity with a lateral size below the resolution limit is not discussed in the manuscript.

Reply: All spatially-resolved experiments have a resolution limit, and the spatial resolution of optical microscopy is approximately given by the wavelength of light (of order 1 micron) – which is small, but is obviously larger than the atomic-scale resolution of a tunneling microscope that can see individual atoms. Happily, our images in Figs. 4 and 5 show clear – and very different— characteristic lengthscales for chiral and Z₃ nematic domains in Co_{1/3}TaS₂ that exceed this resolution limit (a few microns, and 10s-100s of microns, respectively). Moreover, we point out that if the material hosted equal distributions of up- or down- chiral domains (or 0/ 120 /240 degree nematic domains) on lengthscales significantly smaller than our 1 micron spatial resolution, then MCD and MLD images would be expected to show uniformly zero signal, which is clearly not the case.

12. What is the information depth of the used techniques? Can a scenario be ruled out where the magnetic state at the surface is different from the bulk?

Reply: The effective penetration depth of visible light in this material is of order 50-100 nm (based on the material’s conductivity). Experiments of this type therefore effectively constitute a bulk probe, not a surface probe. (Moreover, freshly-exfoliated surfaces showed similar behavior as surfaces exposed for weeks to ambient conditions).

13. What can be concluded for the magnetic coupling between the layers experimentally? Is the proposed 109° coupling in phase I an assumption or a conclusion?

Reply: Firstly, we stress that our manuscript *does not* propose 109 degree coupling in Phase I (as the Reviewer suggests), nor do we propose 109 degree spin order in any other phase. Please see our Reply to the Reviewer’s comment #2 above, where we address this point of potential misunderstanding. Rather, Figures 2 and 3 emphasize that the magnetic configuration in Phase I is a *distorted tetrahedron*. This is a central finding of our work, discussed throughout the main text and Supplemental Material. A key aspect of the spin configuration in Phase I is that it not only possesses chirality, but also breaks in-plane symmetry (ie, has “Z₃ nematicity”). This is a conclusion based on the experimental presence of both MCD and MLD. The balance and interplay of the 4-spin interactions and magnetic anisotropy that can give rise to such a state are discussed in detail in the main text and in the Supplemental Material.

14. Finally, can the proposed uuud state (phase IV) be reproduced in the simulation?

Reply: The proposed uuud state cannot be captured by our Monte Carlo simulations, since we consider a classical spin model. This outcome is expected, as stabilizing this phase requires strong longitudinal

fluctuations that permit unequal moment lengths across the four sublattices. We emphasize that this uuud state is distinct from the conventional *field-induced* uuud phase with half-saturated magnetization, which does emerge in our simulations at very high magnetic fields (>10 T).

Nonetheless, as discussed in the final paragraph of Section II.D of the main text, the proposed uuud state remains the most plausible scenario consistent with the constraints imposed by magnetization, MLD, and MCD measurements. We regard this experimentally-driven inference as a significant outcome of the present study, irrespective of the current limitations of theoretical modeling.

Reply to Reviewer 3

The work by Kirstein et al. presents a comprehensive study of the chiral and nematic states in the antiferromagnet $Co_{1/3}TaS_2$. They mapped out the antiferromagnetic phase diagram under different magnetic fields and temperature ranges using MLD and MCD measurements. A continuous multi- Q manifold model is proposed to understand the phase diagram. Finally, they presented spatially resolved chiral and nematic domains using MCD and MLD microscopy.

I think this paper is well written, and the results are scientifically sound. This work not only demonstrates that $Co_{1/3}TaS_2$ hosts rich nematic and chiral multi- Q states, but also represents a high-quality study of antiferromagnetic textures using optical techniques. I therefore recommend its publication in Nature Communications. Before publication, I have a few comments for the authors to improve the presentation of their manuscript.

Reply: We sincerely appreciate the Reviewer's very positive assessment of the manuscript and recommendation to publish in Nature Communications (after addressing the Reviewer's comments and suggestions for the revised manuscript). We are happy to address and accommodate these suggestions; please see point-by-point reply below.

Comments: 1. Phase I exhibits both MCD and MLD signals. The authors propose that this phase corresponds to a non-equilateral triple- Q state. Could the authors clarify how they rule out the possibility that Phase I arises from the coexistence of spatially separated single- Q and triple- Q domains? In such a case, both MCD and MLD signals could also be expected.

Reply: Excellent question: The spatially-resolved MCD and MLD images indicate that Phase I is not composed of spatially-separated single- Q and triple- Q domains. For example, Fig. 4a shows that large nematic domains are everywhere present in this region (and every other region we measured). That is, to within the experiment's 1-micron spatial resolution, there are no regions without an underlying 0, 120, or 240-degree nematic order. Simultaneously, *imaging the same region* with MCD microscopy shows that domains with non-zero chirality are everywhere present (that is, there are no regions without an underlying positive or negative chirality; see Fig 5). Every spatial region we have studied exhibits both (large) nematic domains and (small) chiral domains in Phase I. Phase I is therefore consistent with a magnetic state possessing both chiral *and* nematic (C_{2z}) order – a distorted triple- Q phase -- rather than spatially separated phases of purely 1 Q and 3 Q order. As another example, Figs. 5b and 5d show that the handedness of the chirality can be uniformly poled (the images are uniformly red and blue, respectively), however when these same regions are imaged using MLD, large domains with nematic (C_{2z}) character completely populate the image (similar to those shown in Fig. 4). The last paragraph of the manuscript does already make this important point, where it states “...*the images confirm that Phase I cannot be a phase-separated mixture of coexisting single- Q and triple- Q ground states*”.

2. Do the authors have an intuitive physical picture or a simple explanation for why the magnetic textures in Phase I exhibit both nematicity and chirality? Although the authors attribute this to magnetic anisotropy, the explanation remains unclear. Magnetic anisotropy is relatively common, but the coexistence of nematic and chiral textures is much less so. What is special about $\text{Co}_{1/3}\text{TaS}_2$ in this context?

Reply: We appreciate the incisive question. The short answer / “intuitive” physical picture is that Phase I arises from the balance between i) the strength of magnetic anisotropy (which favors 1Q order) and ii) the strength of 4-spin interactions that exist in $\text{Co}_{1/3}\text{TaS}_2$ (which favors threefold-symmetric chiral 3Q order). In $\text{Co}_{1/3}\text{TaS}_2$, what is special is that the anisotropy is apparently sufficiently *weak* that it competes effectively with 4-spin interactions, leading to a situation where the magnetic ground state at low temperatures and low fields is noncoplanar triple-Q (which gives chirality) *but* with unequal Fourier components $\Delta_1 \neq \Delta_{2,3}$ (which leads to nematic character as well). Otherwise, if the anisotropy were extremely strong, it would dominate and give single-Q stripe order everywhere (with, say, $\Delta_{2,3}=0$). This interplay between anisotropy and 4-spin interactions in $\text{Co}_{1/3}\text{TaS}_2$ – and how it can lead to the “distorted triple-Q” order in Phase I, is discussed in section II D of the main text (after Equation 3), and in detail in the Supplementary Information (see section 2D, and Figs. S7 and S8 in particular).

3. The MCD curve versus magnetic field in Fig. 2a shows a tilt, suggesting that the field induces some MCD even at high temperatures such as 30 K or 40 K. In contrast, at high temperatures (e.g., 30 K), the MCD is labeled to vanish at large fields in Fig. 2c. The authors should clarify this apparent inconsistency.

Reply: We thank the Reviewer for pointing out an apparent inconsistency. What we intended Fig. 2c to indicate is the presence or absence of any *additional, spontaneous, topological* MCD (due to chiral antiferromagnetic order) that exists on top of the trivial linear-in-field MCD that arises from ordinary Hall effects. **This is now clarified in the Fig. 2c caption of the revised manuscript.**

4. I have some questions regarding Phase IV: (i) In Section A, the authors state that magnetic fields induce two new phases, Phase II and Phase IV. However, this is not evident from Fig. 1. (ii) There is no visible sharp transition in Fig. 2b at high temperatures. How do the authors determine the boundary between Phase III and Phase IV in Fig. 2c?

Reply: Again, we are grateful for the opportunity to clarify an apparent inconsistency. The Reviewer notes that the data in Figs 1c,d *alone* do not show any field-dependence at higher temperatures (27-38 K), and therefore do not explicitly show any phase III-IV boundary. Therefore, where the presence of Phase IV is first mentioned (after discussing Figs. 1c,d), the revised manuscript now clarifies “**(Note that the data shown in Figs. 1c,d do not explicitly reveal the transition to Phase IV; however this transition is shown in Refs. [18, 19, 25], and is also evident in Fig. 2 below).**”

More generally, we also note that the general outlines of the $\text{Co}_{1/3}\text{TaS}_2$ phase diagram – including the presence of high-field Phases II and IV – were recently postulated on the basis of the neutron diffraction, magnetization, electrical transport, and susceptibility studies reported in Refs. [18, 19, 25] (and as our manuscript notes, Figs. 1c and 1d are data from [19]). The primary goal of our present work is to elucidate the underlying nature and magnetic symmetries of these different magnetic phases, using optical methods (we do not lay claim to inventing this notional phase diagram). While the data shown in Fig. 1 alone does not show any field-dependence at high temperatures (27-38 K), and therefore does not show the phase III-IV boundary, we emphasize that our measurements shown in Fig. 2b do. In Fig. 2b, the Phase III-IV boundary is revealed by the steps in MLD vs B, which are indicated by arrows. The steps become broader and less sharp at higher temperatures --as is often the case for magnetic phase transitions at elevated temperatures-- but the position of the steps, accurately determined via the maximum of the

slope $d(\text{MCD})/dB$, allows us to identify this final phase boundary, which in turn is in good agreement with the boundary identified in [18, 19, 25].

5. Figures 4(c) and 4(d) appear repetitive. Moreover, there is no description of Fig. 4(d) in the main text. The authors should address this omission.

Reply: Yes, as the caption states, panel (d) is just the director map of panel (c) overlaid with the first MLD map of panel (a). We agree that perhaps this is a bit repetitive, but during these studies we have found it very helpful and of value to see the two maps directly overlaid, so that different nematic domains can be clearly associated with particular MLD signals (otherwise it's harder to determine whether features in two maps occur at exactly the same location). We anticipate that readers will also find this helpful. And finally, **per the Reviewer's suggestion, the revised manuscript now refers to Fig. 4d in the main text.**

LIST OF CHANGES TO MANUSCRIPT (changes appear in red color in the revised manuscript)

- At the end of section I B where cross-talk effects are first noted and where Fig. S2 is first referenced, we add “*This cross-talk also generates the very small MCD signal between TN1 and TN2 in Fig. 1e (see Supplementary Fig. S2)*” (per Reviewer #1)
- The third sentence of the main text now begins “*Antiferromagnets can exhibit diverse spin configurations...*” (per Reviewer #1)
- Updated Figure 1a, and its caption, now explicitly shows/discusses the different *intra- and inter-layer exchange constants J_1 and J_{c1}* . (per Reviewer #2)
- Throughout the manuscript (including Figures in the main text and Supplement), the terms “equilateral” and “non-equilateral” are replaced with the more explicitly descriptive phrases “*three-fold symmetric*” and “*distorted*” (please note that only the first few instances are highlighted in red color in the revised manuscript). (per Reviewer #2)
- The paper by Haldar et al [PRB **104**, L180404 (2021)], formerly Ref. [38] in the original manuscript but now Ref [26] in the revised manuscript, is now cited in the Introduction section of the revised manuscript, and noted/described more prominently in Section II C “*related AFM phases were studied theoretically for Mn monolayers~\cite{haldar2021}*”, and also in Section II D “*We also note that Ref.~\cite{haldar2021} considered a 1D sub-manifold of the 2D manifold of degenerate ground states..*” (per Reviewer #2)
- The revised manuscript now includes (near the end of Section II D) a description of why magnetic fields favor the threefold-symmetric 3Q state (Phase II), and what energy/anisotropy terms lead to the net residual moment that couples to applied fields. “*The field-induced Phase II is stabilized by the larger residual magnetic moment of the three-fold symmetric triple-Q phase (see the magnetization data in Fig. 1d), favoring a state in which one of the four spin sublattices aligns along \hat{H} . A residual net moment exists in both Phases II and I (even at zero field) due to the anisotropy terms in our model, and its direction ($\pm \hat{c}$) is linked to the sign of the underlying spin chirality.*” (per Reviewer #2)
- The caption of Fig 2c is updated to indicate that the presence/absence of MCD refers to the *spontaneous, topological* MCD (due to chiral antiferromagnetic order) that exists on top of the trivial linear-in-field MCD that arises from ordinary Hall effects. (per Reviewer #3)

- Where Phase IV is first mentioned (after discussing Figs. 1c,d), the revised manuscript now clarifies “(Note that the data shown in Figs. 1c,d do not explicitly reveal the transition to Phase IV; however this transition is shown in Refs. [18, 19, 25], and is also evident in Fig. 2 below).” (per Reviewer #3)
- Figure 4d is now referred to in the main text (per Reviewer #3)
- At the end of Section II A we also explicitly define what we mean by “ C_{2z} ”.

Reply to Reviewers [manuscript NCOMMS-25-30582A, “Tunable chiral and nematic states in the triple-Q antiferromagnet Co_{1/3}TaS₂”]

Apologies to all Reviewers for our delay; travel schedules and experimental neutron beam and magnet time, at Oak Ridge and the Magnet Lab respectively, regrettably delayed our reply. We appreciate the second reviews and the recommendation to publish in Nature Communications from Reviewers #1 and #3, and from the new Reviewer #4. Below we respond in detail to Reviewer #2. For ease of reading, all original remarks from the Reviewers are reprinted in *blue italics*. -Yours sincerely, the authors

Reply to Reviewer #1:

Reviewer #1 (Remarks to the Author): publish as is.

Reply: We appreciate the Reviewer’s second review and recommendation to publish in Nature Comm.

Reply to Reviewer #3:

Reviewer #3 (Remarks to the Author): The authors have satisfactorily addressed all my comments. I do recommend the manuscript for publication in Nature Communications.

Reply: We thank the Reviewer for their second review of our manuscript, and for recommending publication in Nature Communications.

Reply to Reviewer #4:

Reviewer #4 (Remarks to the Author): The revised manuscript is much clearer and addresses the points raised in the first round. For Reviewer 1’s concerns, the phrasing in the Introduction has been adjusted and the small MCD signal between TN₂ and TN₁ is now properly explained as cross-talk.

For Reviewer 2’s main objections about Phase I, the authors have clarified why higher-order exchange interactions cannot explain the data, and why an anisotropy-based model is more consistent with neutron diffraction and magnon gap results. The clarification of the MLD signal behavior is also helpful. The improved definitions and terminology (nematicity, distorted vs three-fold symmetric 3Q) make the presentation stronger.

For Reviewer 3’s questions, the authors have explained why Phase I is not phase coexistence, but a single distorted 3Q state with both chirality and nematicity. The additional discussion of anisotropy, the field-induced phases, and the clarifications in the figures all improve readability.

Overall, the work convincingly demonstrates how combining MCD and MLD can disentangle chiral and nematic orders and provide real-space imaging of magnetic textures. This is a strong contribution, and I recommend acceptance.

Reply: We appreciate Reviewer #4’s positive assessment of our revised manuscript, and for carefully evaluating our detailed responses to the original reviews of Reviewers 1-3. We are delighted that Reviewer #4 finds that we addressed the original questions, concerns, and objections from Reviewers 1, 2, and 3 in a satisfactory manner, and recommends acceptance in Nature Communications.

One small clarification is warranted here, because it comes up again in reviewer #2’s second report: In our original response to Reviewer #2, we discussed why **higher-than-4th-order** (namely, 6th-order) exchange interactions did not capture our data. 4th-order interactions, of the type we already consider in our manuscript (the biquadratic term H_{bq} that is introduced in Eq. 2, in conjunction with single-ion anisotropy and bond-dependent exchange anisotropy of Eq. 3), are a crucial ingredient of our model.

Reply to Reviewer #2:

Reviewer #2 (Remarks to the Author): Despite the substantial and detailed criticism I have raised in the first review round, Kirstein and coworkers have resubmitted a manuscript with only cosmetic changes. In their reply, most of my questions are evaded, some given answers are wrong and the clarifications and explanations which were indeed helpful to understand the work, did unfortunately not make it into the new version of the manuscript.

Reply: We admit to being very puzzled by the Reviewer's point of view, especially since our very detailed (>6 pages) point-by-point response to their original report directly and (we believe) very clearly addressed every one of the Reviewer's criticisms. Re-reading our original response, we maintain that it is comprehensive and accurate (and certainly not "evasive"), and regret that the Reviewer feels otherwise.

We note that all other Reviewers (#1 and #3), as well as the new Reviewer #4 who reviewed the original reports and evaluated our Response, are satisfied with the revised manuscript and Response, and recommend publication. Moreover, it can be easily confirmed that many substantial changes and improvements to the original manuscript were made, in response to the questions and feedback from Reviewer #2 (and #1 and #3). Not every single question or remark from the Reviewers resulted in a change to the manuscript, particularly if the concern was already addressed (or the answer already contained) in the original manuscript; in this case it is sufficient to point it out in the Response.

I do understand the tactics of the rebuttal, but this way of dealing with justified critique is risky, and I remind the authors that two of them have already published a wrong conclusion about the magnetic state of the very same system in Ref. [25].

Reply: Regarding the Reviewer's comment "...and I remind the authors that two of them have already published a wrong conclusion about the magnetic state of the very same system in Ref [25]", we (meaning all the authors of the current manuscript under consideration) would like to make very clear that, in contrast to the Reviewer's implication, the conclusion presented in Ref. [25] was not the result of unjustified assumptions or inconsistent interpretation. Two of us (PP, JGP) made an analysis in Ref. [25] based on the best information available then – specifically, the 1983 neutron diffraction study of Parkin and Brown, which reported a magnetic structure with $\mathbf{Q}=(1/3, 1/3, 0)$, which is a coplanar 120° order. However, subsequent neutron diffraction measurements performed by the authors [PP, JGP, in Ref. 19] and by others [Takagi, Ref. 18], revealed a fundamentally different non-coplanar magnetic order characterized by $\mathbf{Q}=(1/2, 0, 0)$, which naturally requires an updated explanation (triple- \mathbf{Q} order) based on this new observation. Therefore, the discrepancy arises from the discovery that the original reference data used in Ref. [25] turned out to be inconsistent. Unless the Reviewer is suggesting that we, as scientists, are generally expected to automatically doubt the validity of peer-reviewed experimental results, this criticism is potentially misleading and, we feel, quite unfair and misplaced.

Therefore, most of my critique is still valid and I come to the same conclusion, that I do not see substantially new physics within the manuscript which is conclusively backed up by experiment to warrant publication in Nature Communication. In my opinion, certain inconsistencies of the manuscript need to be addressed before publication in any journal. Whether these problems can be fixed by simply writing more clearly or point toward deeper flaws of the work, I do not know.

Reply: We are sorry to hear this view; we would like to note that all three other Reviewers (#1, #2, and #4) were satisfied with our revised manuscript and recommended it for publication. Below we respond, point-by-point, to the Reviewer's second report.

For the future, I encourage the authors to refrain from the overuse of italic, bold and underlining in the

review correspondence. Reviewers are quite capable of reading texts carefully without such "help".

Reply: OK.

In the following, I will not again comment on all problems of the manuscript and shortcomings of the rebuttal, but only address the ones I deem most critical.

1. Let us start with this claim from the rebuttal: "Moreover, we respectfully emphasize that the new findings include not only the experimental identification of a distorted triple-Q antiferromagnetic phase with co-existing chiral and C_{2z} symmetries (noted by the Reviewer), but also 1) the demonstration that magneto-optical methods for circular and linear dichroism can function as incisive reporters of these antiferromagnetic symmetries, ..."

At face value, this sentence is circular: Fundamentally, you cannot establish a new magnetic technique using a specific spin texture and then also claim that the observed spin texture is a new finding. Now, it seems that CoTa3S6 has four different magnetic phases: which spin textures are used to calibrate MLD and MCD and which results are actually new?

Reply: We are having difficulty understanding this concern. Hopefully the Reviewer does not think we are claiming the invention of circular dichroism (CD) and linear dichroism (LD) methods, for detecting Hall conductivity ($\sim \sigma_{xy}$) and in-plane anisotropy ($\sim \sigma_{xx} - \sigma_{yy}$), respectively. Definitely, we are not! Both LD and CD are well-established optical techniques that have been used for over a century: LD is famously sensitive to order parameters that break in-plane symmetry (such as crystal anisotropy, or nematicity), whereas CD is an established reporter of time-reversal-symmetry breaking magnetic phenomena that lead to nonzero σ_{xy} (such as ferromagnetism, or the chiral antiferromagnetism we consider). Crucially, LD was also recently demonstrated to be an incisive probe of three-state magnetic nematicity resulting from stripe-like (1Q) order, in the closely-related triangular and hexagonal antiferromagnets $\text{Fe}_{1/3}\text{NbS}_2$, FePS_3 , and FePSe_3 [e.g., *Nature Materials* **19**, 1062 (2020); *Nano Letters* **21**, 6938 (2021); *Nature Physics* **20**, 1888 (2024)]. Moreover, CD was shown to detect nontrivial antiferromagnetic orders that generate σ_{xy} [e.g., in Mn_3Sn : *Nature Photonics* **12**, 73 (2018); *Appl. Phys. Lett.* **114**, 032401 (2019)]. All these papers are cited in our manuscript.

Our work applies these methods to the case of $\text{Co}_{1/3}\text{TaS}_2$, where the existence of four different competing antiferromagnetic phases was recently postulated (e.g., Takagi, Ref. 18), but the underlying magnetic symmetries of these four phases were unknown or incorrectly identified. The significant advance our work makes is the new surprising finding and classification of these various phases based on the presence or absence of CD and LD – that is, by the presence or absence of Hall conductivity (σ_{xy}) and in-plane anisotropy ($\sigma_{xx} - \sigma_{yy}$), as shown in Figure 2. Let us state that these findings were never reported before and are very new. Together with neutron, transport, and magnetization data, and supported by theoretical modeling, this allows us to identify Phase III as a purely single-Q phase (only nematicity, no chirality), Phase II as a threefold-symmetric triple-Q phase (only chirality, no nematicity), Phase I as a distorted triple-Q phase (possessing both chirality and nematicity), and Phase VI as having neither chiral nor nematic character. This is a central new aspect of the work, not achieved previously. We note that the magnetic structure of Phase I was incorrectly identified in Ref. [18] as being purely chiral, likely because Ref. [18] was unable to test for nematicity. We also note that Phase III was independently and very recently suggested to have purely single-Q character, based on inelastic neutron scattering studies of the magnetic excitation spectrum [*Phys. Rev. X* **15**, 031032 (2025)], consistent with our findings.

We do not see any circular logic here – we used well-established optical probes of underlying symmetry, applied them to an interesting new antiferromagnet, and have revealed the presence/absence of underlying

symmetries. Together with other data and modeling, this allows us to identify the nature of the various magnetic states in these four phases, which had not previously been achieved. Moreover, being optical methods, we have probed the various antiferromagnetic orders with micron spatial resolution to directly image the chiral and nematic domains, which also had not previously been achieved. Finally, this work demonstrates that in this class of antiferromagnets with chiral and/or nematic character, CD and LD function as incisive and flexible probes of the underlying magnetic order. We anticipate that many people working in the broad fields of magnetism and optics will be very pleased (and likely surprised) to learn that these optical methods can be used to study certain classes of antiferromagnets, because historically antiferromagnetism has evaded detection by such linear-optical means. This work suggests that the full power of spectrally-, spatially-, and temporally-resolved optical techniques can be brought to bear on such antiferromagnets, echoing decades of similar progress in optical studies of ferromagnets.

2. Another problem puts a main result of the work into question. Here, again a sentence from the rebuttal: "Finally, we note that the MLD signal (a reporter of 1Q order parameter) emerges and grows throughout Phase III, achieving a maximum at the Phase I – Phase III boundary, which remains approximately constant as the sample is cooled throughout Phase I (i.e., the MLD is not larger in Phase I as the Reviewer suggests). Figure 1g shows this reasonably well, and Supplemental Fig. S2d shows this extremely clearly."

This description is wrong! In both plots the highest MLD signal is found at the lowest temperature of $T=4.2$ K, not at $T=26.5$ K. But let us assume for the sake of argument, that the MLD signal is constant in phase I; even then the problem arises that the MLD signal interpreted as a direct measure of 1Q character is INCONSISTENT with the interpretation of phase III as a pure 1Q state and phase I as a mixed 1Q-3Q state. When the state becomes more 3Q-like in phase I, the 1Q character needs to drop. There seems to be a further temperature dependence which is not discussed and not accounted for.

Reply: Below we reprint Fig. 1g and Fig. S2d from the manuscript. As stated in our original response, the experimentally-measured linear dichroism grows throughout Phase III ($T=37$ to 26.5 K)... and remains approximately constant as the sample is cooled throughout Phase I ($T<26.5$ K). We defer to the Editors and other Reviewers whether the exactness of our statement is actually an important point.

Regardless, this all seems somewhat beside the point in our view, because there is no law or requirement that the linear dichroism signal – a reporter of broken in-plane symmetry and nematic character – must drop (or change at all) upon transitioning from Phase III (single-Q) to Phase I (distorted triple-Q) at finite temperature. (Note: this would not be true at $T=0$ where order parameters are maximal and saturated; perhaps the Reviewer has the $T=0$ case in mind). At $T=26$ K (and below), the amplitudes of the order parameters are still growing. Linear dichroism (LD) is simply a reporter of broken C_3 symmetry, and will therefore be non-zero if the amplitudes of the three single-Q wave vectors ($\Delta_1, \Delta_2, \Delta_3$) are unequal -- for details see Equation 1, and Fig. 3. Likely, the LD signal scales with their difference [e.g., as something like $\Delta_1^2 - (\Delta_2^2 + \Delta_3^2)/2$, although the actual relationship is not known]. If Δ_1, Δ_2 , and Δ_3 grow with decreasing temperature approximately following the notional sketch below (left plot), then the LD signal will indeed grow throughout Phase III and then remain approximately constant upon cooling into Phase I.

Crucially, we emphasize that such a temperature dependence of Δ_1 is, in fact, already strongly suggested and motivated by the experimentally-measured neutron diffraction intensity that is shown in Fig. 1c of the main text (right plot). The antiferromagnetic diffraction intensities should scale with $|\Delta_1|^2$. Thus, the measured LD – a measure of nematicity – is not required to drop when the magnetic order parameter acquires some triple-Q character upon cooling through $T=26.5\text{K}$ and entering Phase I.

We do find it very interesting that the emergence and growth of the distorted triple-Q phase (Phase I) does seem to approximately maintain its degree of nematicity (as measured by MLD), but at the moment we do not know whether this is coincidence or points to something deeper.

3. As an answer to my comment #4 [page 2, "coexistence of single-Q and triple-Q orderings within the same system – an uncommon feature in real materials.": Firstly, a distorted 3Q state has been proposed already in Ref. 38. Secondly, 1Q, 3Q and transient 2Q states co-exist in the Mn/Re(0001) system due to different stacking order and inside domain walls, respectively] the authors write:

"As the authors of [38] will surely agree, such an unusual distorted 3Q state is indeed not common – we carefully searched through the existing literature for relevant precedent, and found that Mn/Re was one of only a very few examples containing related physics."

This is a very misleading statement, and the term "as the authors will surely agree" is pure speculation. There are only a handful of papers out there dealing with 3Q states at all (in the sense of nearest neighbor spins having angles close to 109.5° , not in the sense of skyrmion lattices and such). As far as I know, the first experimental observation was in the Mn/Re(0001) system by SP-STM, where 1Q and 3Q areas co-exist (due to different possible stacking order of the Mn layer) and in addition the 3Q state itself might be distorted (Haldar et al.). The former work is not even cited, although it is quite relevant for the discussion.

One cannot say whether something is common or uncommon if there are only a handful of cases in total. Furthermore, one might say that the ideal 109.5° -3Q state will never be observed in real systems, because there are always tiny interactions present which distort it.

Reply: This debate (see original Reviews and our Response) seems to center on a definition of the word “uncommon”. To avoid any misunderstanding, we’ve removed the phrase “...an uncommon feature in real materials” from the Introduction section of the manuscript, with no loss of meaning or generality.

Following the Reviewer’s comment, we also added the experimental work as a new reference.

4. The limited spatial resolution of MLD and MCD is an issue which is unfortunately not openly discussed in the manuscript. For a single image data point, the information seems to come from a volume 50-100 nm in depth and about 1000 nm in diameter. I agree with referee #3 that the authors do not convincingly exclude a scenario for phase I where separate 1Q and 3Q domains co-exist beyond the resolution limit, compared to the proposed laterally homogenous and itself distorted 3Q state (with 1Q component).

Reply: We are puzzled why the Reviewer claims that the “*the limited spatial resolution of MLD and MCD is an issue which is unfortunately not openly discussed in the manuscript*”. The spatial resolution of our microscope (approx. 1 micrometer) is very explicitly stated in the Methods (Section IV.B). This is close to the fundamental diffraction limit of red/infrared light (as is the case for most optical microscopy studies with decent optics). Optical microscopy obviously cannot probe what is happening at nanometer or atomic length scales. But we can, and do, say that the images of Fig. 4 and S5 quite clearly show large hundreds-of-micron lengthscale spatial patterns having discrete C_{2z} (nematic) character, suggesting large domains with nematicity. Moreover, the images in Fig. 5 also show, in the exact same regions, smaller few-micron scale patterns and domains having chiral character. As we have explained in detail (please see our original response to Reviewer #3, which we hope the Reviewer read), these experimental observations, together with the fact that the images exhibit uniform chiral character when poled in tiny magnetic fields (Fig. 5b,d), are consistent with a laterally-homogeneous distorted triple-Q state.

To restate our reasoning more clearly: if Phase I truly consisted of separate 1Q and 3Q domains, then the resulting nematic domain pattern—extending over several hundred micrometers—should have appeared in the MCD images as regions with void signals. However, no such features are observed.

We note that both Reviewers #3 and #4 are satisfied with our response to Reviewer #3’s related question about co-existing 1Q and 3Q domains. Nonetheless, to make it perfectly clear that optical microscopy does not probe below the diffraction limit of light, we amend the penultimate sentence of the manuscript’s last paragraph to say “*These images confirm that, to within our microscope’s 1 micron spatial resolution, Phase I is not a phase-separated mixture of coexisting single-Q and triple-Q ground states*”.

5. Concerning the proposed states for phase I and II, the comments in the rebuttal were quite helpful. I try to be brief here, but I think the present manuscript is not clear enough at this point. The authors need to clearly discriminate at least these three states: 1) the ideal, fully symmetric 109.5°-3Q state, 2) the state with one spin up and three spins symmetrically distorted, sometimes called the “threefold symmetric 3Q” and 3) the 3Q state which is distorted by crystal anisotropy, which looks like a 2Q state seen from the top. There might be more. In any case, I was confused, because the authors mark the central (red) point on the manifold in Fig. 3a as the three-fold symmetric 3Q. But at this point on the manifold is the ideal 109.5°-3Q. As far as I understand, the state proposed for phase II is not even on the manifold, because it has a higher Heisenberg energy than the ideal 109.5°-3Q.

Reply: We thank the Reviewer for raising this point which, we now recognize, will benefit from additional clarification in the paper. We agree that a clearer distinction is needed especially between case (2) and an idealized fully symmetric configuration with 109.5° angles. The revised and updated manuscript now explicitly differentiates:

- (1) **Ideal three-fold symmetric 3Q state.** This state has equal Fourier amplitudes ($\Delta_1 = \Delta_2 = \Delta_3$), mutual spin angles of 109.5°, and zero net magnetization. This state preserves three-fold rotational symmetry and corresponds to the red central point on the multi-Q manifold in Fig. 3a.
- (2) **Canted three-fold symmetric 3Q state (Phase II).** This state also has $\Delta_1 = \Delta_2 = \Delta_3$ and therefore retains three-fold rotational symmetry, but possesses an additional small net magnetization due to field-induced (and/or anisotropy-induced) canting. This net moment is readily described by including a small ferromagnetic Fourier component \mathbf{S}_0 to the three-dimensional manifold of Equation (1), and is already captured by our model (see Supplemental Equation S7). This canted state is the Phase II ground state when a finite field or bond-dependent anisotropic exchange term $H_{\pm\pm}$ (which corresponds to the “anisotropic symmetric exchange” mentioned by the Reviewer in the following comment) is included.

(3) **Distorted 3Q state (Phase I).** This state breaks three-fold symmetry through unequal Fourier amplitudes ($\Delta_1 > \Delta_2 = \Delta_3$), giving rise to nematicity (as well as chirality).

We emphasize that cases (1) and (2) share a key property: both maintain $\Delta_1 = \Delta_2 = \Delta_3$ and preserve three-fold symmetry, and are therefore chiral but not nematic, regardless of the precise values of their mutual spin angles. In this study, the presence or absence of three-fold symmetry –and hence nematicity, as measured by linear dichroism– is the primary focus of distinguishing criterion. We also stress that our theoretical framework already systematically captures the evolution from case (1) to case (2) through the development of a net magnetic moment $\tilde{\mathbf{S}}_0$; please see Eq. S7 in the Supplementary Information.

We agree that this is a useful distinction, and the revised manuscript now includes additional figures that explicitly distinguish cases (1) and (2) (see new Fig. S6d and its caption), and provides a more detailed discussion in the main text:

(Page 4) *“The case $\Delta_1 = \Delta_2 = \Delta_3$ (red circle in Fig. 3a) yields a chiral and three-fold symmetric state, where the four spin sublattices align along the principal axes of a regular tetrahedron with 109.5° mutual angles. Phase II corresponds to this case, but with the addition of a small net out-of-plane magnetization arising from spin canting induced by the applied magnetic field. This canted configuration is naturally described by including a constant (ferromagnetic) Fourier component $\tilde{\mathbf{S}}_0$ to the three-dimensional manifold in Equation (1); for additional details see Supplemental Note II.A. (Equation S7). Importantly, even with canting, the three-fold symmetry remains intact (see Fig. S6d) and therefore we refer to Phase II as a “three-fold symmetric triple-Q” state through this work, to emphasize its rotational symmetry (though not implying a perfect 109.5° tetrahedral configuration.)”*

Two more comments: starting from the ideal 109.5° -3Q state, uniaxial anisotropy alone does not produce a net moment. Secondly, the authors might want to consider the anisotropic symmetric exchange interaction, which can effectively couple the 3Q state in the way they propose for phase II (1 spin up), see F. Nickel et al., new Ref. [51].

Reply: We appreciate the Reviewer’s suggestion, and we agree that uniaxial anisotropy alone does not produce a net moment (our manuscript does not claim this). And we agree that anisotropic symmetric exchange can do so – in fact our model already does incorporate a term along these lines, as suggested by Nickel et al [Ref 51]; this is precisely the “bond-dependent exchange anisotropy” term, $H_{\pm\pm}$, that was introduced in Equation (3) of the main text (and discussed in more detail in the Supplement).

6. Finally, as the authors are aware, higher-order interactions (HOI) such as the bi-quadratic one are needed to get a non-coplanar state such as the 3Q state, because an interaction is needed which favors 90° angles. I have not found any reference or mention of HOIs in the main text of the manuscript. This is odd, because without HOI neither phase I and phase II, nor the T-driven transition from phase I to phase III can be understood.

Reply: Higher-order interactions of the type noted by the Reviewer (e.g., bi-quadratic) are already incorporated into our model from the very beginning. We refer the Reviewer to section II.D. of the manuscript, Equations (2) and (3) in particular. Equation (2) and its preceding paragraph explicitly introduce the bi-quadratic term H_{bq} that is a crucial ingredient of our model. As the reviewer notes, this bi-quadratic term is a higher-order (four-spin) interaction, although the text does not explicitly refer to it as such. **The revised manuscript now does.** See also Supplemental Section II.B., which is entitled “Four-spin interactions”. Together with H_{bq} , the single-ion anisotropy term H_{SI} and the bond-dependent exchange anisotropy $H_{\pm\pm}$ (both defined in Equation 3) successfully capture the phase transitions between phase I and II, and the temperature-driven phase transitions between phase I and III – as already shown in Figure 3f and 3g of the main text and Figs. S7 and S8 of the Supplementary Material.

Changes to manuscript (these new updates appear in orange color in the revised manuscript):

Amend the penultimate sentence of the manuscript's last paragraph to say “*These images confirm that, to within our microscope's 1 micron spatial resolution, Phase I is not a phase-separated mixture of coexisting single-Q and triple-Q ground states*”.

To avoid any potential misinterpretation, we've removed the phrase “*...an uncommon feature in real materials*” from the Introduction section of the manuscript, with no loss of generality.

Sentence preceding Equation (2): Add “...we find that a simple real-space biquadratic (i.e., four-spin) term H_{bq} captures...”. And in the sentence preceding Supplemental Equation (S1): Add “...that includes bilinear and higher-order four-spin interactions:”

Added a new citation: J. Spethmann, S. Meyer, K. von Bergmann, R. Wiesendanger, S. Heinze, A. Kubetzka, Discovery of Magnetic Single- and Triple-q States in Mn/Re(0001), Phys. Rev. Lett. **124**, 227203 (2020).

New Supplemental Figure S6d (and caption), which explicitly compares and discusses the threefold symmetric 3Q ‘canted’ spin configuration of Phase II with the idealized and perfectly tetrahedral spin configuration.

Main text, page 4, Section II.D., second paragraph is updated to read: “*The case $\Delta_1 = \Delta_2 = \Delta_3$ (red circle in Fig. 3a) yields a chiral and three-fold symmetric state, where the four spin sublattices align along the principal axes of a regular tetrahedron with 109.5° mutual angles. Phase II corresponds to this case, but with the addition of a small net out-of-plane magnetization arising from spin canting induced by the applied magnetic field. This canted configuration is naturally described by including a constant (ferromagnetic) Fourier component \tilde{S}_0 to the three-dimensional manifold in Equation (1); for additional details see Supplemental Note II.A. (Equation S7). Importantly, even with canting, the three-fold symmetry remains intact (see Fig. S6d) and therefore we refer to Phase II as a “three-fold symmetric triple-Q” state through this work, to emphasize its rotational symmetry (though not implying a perfect 109.5° tetrahedral configuration.)*”